# PARAMETRIC UMAP: LEARNING EMBEDDINGS WITH DEEP NEURAL NETWORKS FOR REPRESENTATION AND SEMI-SUPERVISED LEARNING

## ABSTRACT

We propose Parametric UMAP, a parametric variation of the UMAP (Uniform Manifold Approximation and Projection) algorithm. UMAP is a non-parametric graph-based dimensionality reduction algorithm using applied Riemannian geometry and algebraic topology to find low-dimensional embeddings of structured data. The UMAP algorithm consists of two steps: (1) Compute a graphical representation of a dataset (fuzzy simplicial complex), and (2) Through stochastic gradient descent, optimize a low-dimensional embedding of the graph. Here, we replace the second step of UMAP with a deep neural network that learns a parametric relationship between data and embedding. We demonstrate that our method performs similarly to its non-parametric counterpart while conferring the benefit of a learned parametric mapping (e.g. fast online embeddings for new data). We then show that UMAP loss can be extended to arbitrary deep learning applications, for example constraining the latent distribution of autoencoders, and improving classifier accuracy for semi-supervised learning by capturing structure in unlabeled data.[1]

Current non-linear dimensionality reduction algorithms can be divided broadly into non-parametric algorithms which rely on the efficient computation of probabilistic relationships from neighborhood graphs to extract structure in large datasets (e.g. UMAP (McInnes et al., 2018), t-SNE (van der Maaten & Hinton, 2008), LargeVis (Tang et al., 2016)), and parametric algorithms, which, driven by advances in deep-learning, optimize an objective function related to capturing structure in a dataset over neural network weights (e.g. Hinton & Salakhutdinov 2006; Ding et al. 2018; Ding & Regev 2019; Szubert et al. 2019; Kingma & Welling 2013).

The goal of this paper is to wed those two classes of methods: learning a structured graphical representation of the data and using a deep neural network to embed that graph. Over the past decade several varients of the t-SNE algorithm have proposed parameterized forms of t-SNE (Van Der Maaten, 2009; Gisbrecht et al., 2015; Bunte et al., 2012; Gisbrecht et al., 2012). In particular, Parametric t-SNE (Van Der Maaten, 2009) performs exactly that wedding; training a deep neural network to minimize loss over a t-SNE graph. However, the t-SNE loss function itself is not well suited to be optimized over deep neural networks using contemporary training schemes. In particular, t-SNE's optimization requires normalization over the entire dataset at each step of optimization, making batch-based optimization and on-line learning of large datasets difficult. In contrast, UMAP is optimized using negative sampling (Mikolov et al., 2013; Tang et al., 2016) and requires no normalization step, making it more well-suited to deep learning applications. Our proposed method, Parametric UMAP, brings the non-parametric graph-based dimensionality reduction algorithm UMAP into a emerging class of parametric topologically-inspired embedding algorithms (Reviewed in A.5).

In the following section we broadly outline the algorithm underlying UMAP to explain why our proposed algorithm, Parametric UMAP, is particularly well suited to deep learning applications. We contextualize our discussion of UMAP in t-SNE, to outline the advantages that UMAP confers over t-SNE in the domain of parametric neural-network based embedding. We then perform experiments

---

[1]Google Colab walkthrough

comparing our algorithm, Parametric UMAP, to parametric and non-parametric algorithms. Finally, we show a novel extension of Parametric UMAP to semisupervised learning.

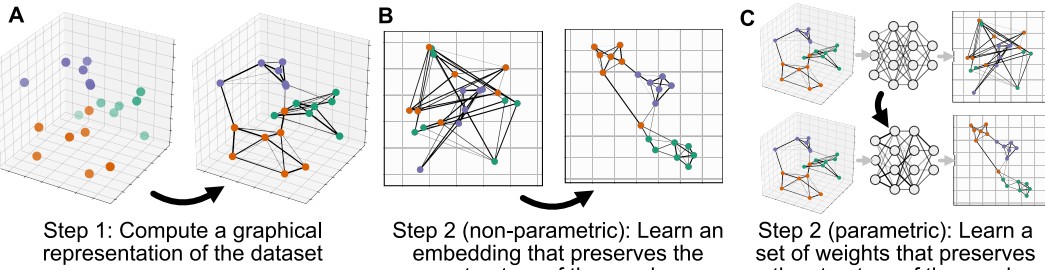

Figure 1: Overview of UMAP (A → B) and Parametric UMAP (A → C). (A) The first stage of the UMAP algorithm is to compute a probabilistic graphical representation of the data. (B) The second stage of the UMAP algorithm is to optimize a set of embeddings to preserve the structure of the fuzzy simplicial complex. (C) The second stage of UMAP, learning a set of embeddings which preserves the structure of the graph, is replaced with a neural network which learns a set of neural network weights (parameters) that maps the high-dimensional data to an embedding. Both B and C are learned through the same loss function.

## 0.1 PARAMETRIC UMAP

t-SNE and its more recent cousin UMAP are both non-parametric graph-based dimensionality reduction algorithms that learn embeddings based upon local structure in data. Much prior work has focused on extending t-SNE to learn the mapping between data and embeddings, for example optimizing t-SNE's loss over learned neural network weights (Van Der Maaten, 2009). However, t-SNE optimizes it's embeddings in a manner poorly suited to being translated to neural network optimization, requiring significant modification of the learning algorithm to train via stochastic gradient descent. Conversely, UMAP's loss is optimized in a manner that can be directly translated for neural network optimization. Here, we explain UMAP's compatability with deep learning applications, why this differs for t-SNE, and how we take advantage. A full discussion of UMAP and t-SNE explaining this contrast is given in the Appendix A.1.

To summarize, both t-SNE and UMAP rely on the construction of a graph, and a subsequent embedding that preserves the structure of that graph (Fig. 1). UMAP learns an embedding by minimizing cross entropy sampled over positively weighted edges (attraction), and using negative sampling randomly over the dataset (repulsion), allowing minimization to occur over sampled batches of the dataset. t-SNE, meanwhile, minimizes a KL divergence loss function normalized over the entire set of embeddings in the dataset using different approximation techniques to compute attractive and repulsive forces.

Because t-SNE optimization requires normalization over the distribution of embedding in projection space, gradient descent can only be performed after computing edge probabilities over the entire dataset. Projecting an entire dataset into a neural network between each gradient descent step would be too computationally expensive to optimize however. The trick that Parametric t-SNE proposes to this problem is to split the dataset up into large batches (e.g. 5000 datapoints in the original paper) that are independently normalized over and used constantly throughout training, rather than being randomized. Conversely, UMAP can be trained on batch sizes as small as a single edge, making it suitable for minibatch training needed for memory-expensive neural networks trained on large datasets as well as on-line learning.

Given these design features, the UMAP algorithm is better suited to deep neural networks, and is more extendable to typical neural network training regimes. Parametric UMAP can be defined simply by applying the UMAP cost function to a deep neural network over mini batches using negative sampling within minibatches. In our implementation, we keep all hyperparameters the

same as the original UMAP implementation, and optimize over the UMAP loss function using the Adam (Kingma & Ba, 2014) optimizer[2].

# 1 UMAP AS A REGULARIZATION

In machine learning, regularization refers to a modification of the learning algorithm to improve generalization to new data. Here, we consider both regularizing neural networks with UMAP loss, as well as using additional loss functions to regularize UMAP.

While non-parametric UMAP optimizes UMAP loss directly over embeddings (Fig 2A), our proposed algorithm, Parametric UMAP, applies the same cost function over an encoder network (Fig 2B). Beyond the parameters being optimized, the optimization steps are equivalent. Given this simple extension toward optimizing over a neural network, UMAP can further be extended to arbitrary neural network architectures and loss functions. Specifically, we use UMAP loss over unlabeled data as a regularization term for a classifier network trained on labeled data (semi-supervised learning), as well as an autoencoder loss as an additional regularization for UMAP latent projections. In the experiments section, we quantitatively explore these networks.

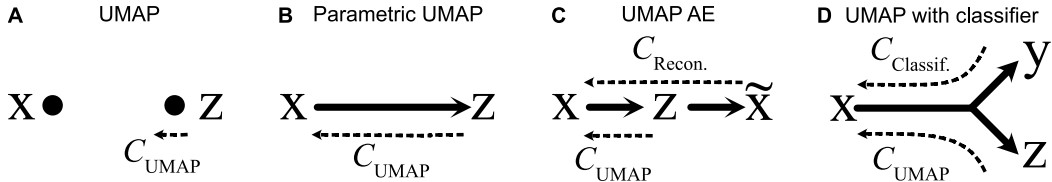

Figure 2: An outline of the varients of UMAP used in this paper. Solid lines represent neural networks. Dashed lines represent error gradients.

## 1.1 AUTOENCODING WITH UMAP

AEs are by themselves a powerful dimensionality reduction algorithm (Hinton & Salakhutdinov, 2006). Thus, combining them with UMAP may yield additional benefits in capturing latent structure. We used an autoencoder as an additional regularization to Parametric UMAP (Fig 2C). A UMAP/AE hybrid is simply the combination of the UMAP loss and a reconstruction loss and apply both over the network. VAEs have similarly been used in conjunction with Parametric t-SNE for capturing structure in animal behavioral data (Graving & Couzin, 2020) and combining t-SNE, which similarly emphasizes local structure, with AEs aids in capturing more global structure over the dataset (van der Maaten & Hinton, 2008; Graving & Couzin, 2020).

## 1.2 SEMISUPERVISED LEARNING

UMAP can also be used to regularize supervised classifier networks, training the network on a combination of labeled data with the classifier loss and unlabeled data with the UMAP loss (Fig 2D). Semi-supervised learning refers to the use of unlabeled data to jointly learn the structure of a dataset while labeled data is used to optimize the supervised objective function, such as classifying images. Here, we explore how UMAP can be jointly trained as an objective function in a deep neural network alongside a classifier.

In the example in Fig 3, we show an intuitive example of semisupervised learning using UMAP over the Moons dataset (Pedregosa et al., 2011). By training a Y-shaped network (Fig 2D) both on the classifier loss over labeled datapoints (Fig 3A, red and blue) and the UMAP loss over unlabeled datapoints (Fig 3A, grey) jointly, the shared latent space between between the UMAP and classifier network pulls apart the two moons (Fig 3B), resulting a a decision boundary that divides cleanly between the two distributions in dataspace.

---

[2]See code implementations: Experiments (www.placeholder.com) Python package (www.placeholder.com)

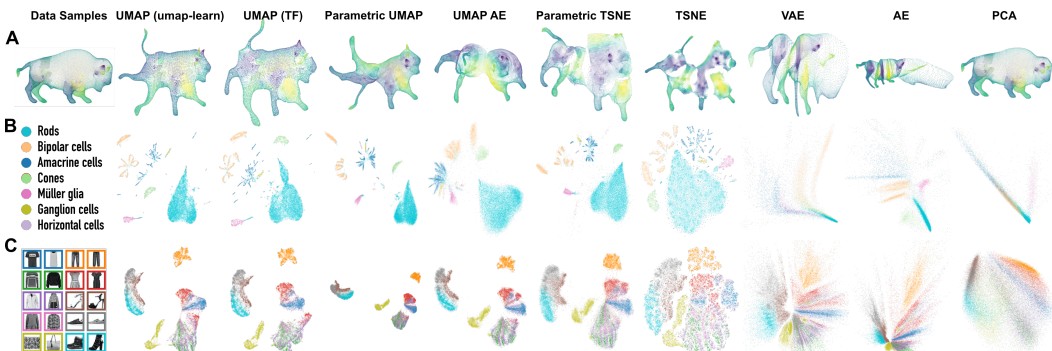

Figure 3: An example of semi-supervised learning with UMAP on the moons dataset. (A) A decision contour learned over the moons dataset with 3 labeled datapoints from each class. Unlabled datapoints are shown in grey and labeled datapoints are shown in red and blue. The decision contour is shown in the background using the 'coolwarm' colormap. (B) The learned embeddings in the jointly trained network. (C) UMAP loss over the unlabeled training dataset. (D) Classifier accuracy for the training and validation set.

## 2 EXPERIMENTS

Figure 4: Example comparison of 2D projections from Parametric UMAP and each baseline. (A) 3D buffalo. (B) Mouse retina single cell transcriptomes. (C) Fashion MNIST. 2D projections of the remaining datasets are given in Fig. 12.

Experiments were performed comparing our novel algorithms, Parametric UMAP and a UMAP/AE hybrid, to several baselines: nonparametric UMAP, nonparametric t-SNE (FIt-SNE) (Poličar et al., 2019), Parametric t-SNE, an AE, a VAE, and a PCA projections. Many additional relevant comparisons (i.e. those discussed in A.5) are not made, as those methods had already been compared to non-parametric UMAP, and our embedding comparisons were performed to confirm similar quality to their non-parametric counterparts, not to outperform them. We additionally compare a second non-parametric UMAP implementation that has the same underlying code as Parametric UMAP, but where optimization is performed over embeddings directly, rather than neural network weights. This comparison is made to provide a bridge between the UMAP-learn implementation and parametric UMAP, to control for variation exogenous to differences parametric versus non-parametric embedding. Parametric t-SNE, Parametric UMAP, the AE, VAE, and the UMAP/AE hybrid use the same neural network architectures and optimizers within each dataset (described in A.2 and A.3). We used the common machine learning benchmark datasets MNIST, FMNIST, and CIFAR10 alongside two real-world datasets in areas where UMAP has proven a useful tool for dimensionality reduction: a single-cell retinal transcriptome dataset (Macosko et al., 2015), and a bioacoustic dataset of Cassin's vireo song, recorded in the Sierra Nevada mountains (Hedley, 2016a;b).

### 2.1 EMBEDDINGS

We first confirm that Parametric UMAP produces embeddings that are of a similar quality to nonparametric UMAP. To quantitatively measure the quality of embeddings we compared embedding algorithms on several metrics across datasets. We compared each method/dataset on 2D and 64D

projections[3]. Each metric is explained in detail in A.4. The 2D projection of each dataset/method is shown in Fig 4. The results are given in Figs 13-17 and Tables 2-6, and summarized below.

**Trustworthiness** Trustworthiness (Eq. 5, Venna & Kaski 2006) is a measure of how much of the local structure of a dataset is preserved in a set of embeddings. In 2D, we observe each of the UMAP algorithms performs similarly in trustworthiness, with t-SNE being slightly more trustworthy in each dataset (Fig 13; Table 2). At 64D, PCA, AE, VAE, and Parametric t-SNE are most trustworthy in comparison to each UMAP implementation, possibly reflecting the more approximate repulsion (negative sampling) used by UMAP.

**KNN-Classifier** A KNN-classifier is used as a baseline to measure supervised classification accuracy based upon local relationships in embeddings. We find KNN-classifier performance largely reflects trustworthiness (Figs 14, 15; Tables 3m 4). In 2D, we observe a broadly similar performance between UMAP and t-SNE varients, each of which is substantially better than the PCA, AE, or VAE projections. At 64 dimensions UMAP projections are similar but in some datasets (FMNIST, CIFAR10) slightly underperform PCA, AE, VAE, and Parametric t-SNE .

**Silhouette score** Silhouette score measures how clustered a set of embeddings are given ground truth labels. In 2D, across datasets, we tend to see a better silhouette score for UMAP and Parametric UMAP projections than t-SNE and Parametric t-SNE, which are in turn more clustered than PCA in all cases but CIFAR10, which shows little difference from PCA (Fig 16; 5). The clustering of each dataset can also be observed in Fig 4, where t-SNE and Parametric t-SNE are more spread out within cluster than UMAP. In 64D projections, we find the silhouette score of Parametric t-SNE is near or below that of PCA, which are lower than UMAP-based methods. We note, however, that the poor performance of Parametric t-SNE may reflect setting the degrees-of-freedom ($\alpha$) at $d-1$ which is only one of three parameterization schemes that Van Der Maaten (2009) suggests. A learned degrees-of-freedom parameter might improve performance for parametric t-SNE at higher dimensions.

**Clustering** To compare clustering directly across embeddings, we performed $k$-Means clustering over each latent projection and compared each embedding's clustering on the basis of the normalized mutual information (NMI) between clustering schemes (Fig 17; Table 6). In both the 2D and 64D projections, we find that NMI corresponds closely to the silhouette score. UMAP and t-SNE show comperable clustering in 2D, both well above PCA in most datasets. At 64D, each UMAP approach shows superior performance over t-SNE.

## 2.2 SPEED

**Training speed** Optimization in non-parametric UMAP is not influenced by the dimensionality of the original dataset; the dimensionality of the dataset only comes into play in computing the nearest-neighbors graph. In contrast, training speeds for Parametric UMAP are variable based upon the dimensionality of data and the architecture of the neural network used. The dimensionality of the embedding does not have a substantial effect on speed. In Fig 5, we show the cross-entropy loss over time for Parametric and Non-parametric UMAP, for the MNIST, Fashion MNIST, and Retina datasets. Across each dataset we find that non-parametric UMAP reaches a lower loss more quickly than Parametric UMAP, but that Parametric UMAP reaches a similar cross-entropy within an order of magnitude. Thus, Parametric UMAP can train more slowly than non-parametric UMAP, but training times remain within a similar range making Paramatric UMAP reasonable alternative to non-parametric UMAP in terms of training time.

**Embedding and reconstruction speed** A parametric mapping allows embeddings to be inferred directly from data, resulting in a quicker embedding than non-parametric methods. The speed of embedding is especially important in signal-processing paradigms where near real-time embedding speeds are necessary. For example in brain-machine interfacing, bioacoustics, and computational

---

[3]Where possible. In contrast with UMAP, Parametric UMAP, and Parametric t-SNE, Barnes Huts t-SNE can only embed in two or three dimensions (Van Der Maaten, 2014) and while FIt-SNE can in principle scale to higher dimensions (Linderman et al., 2019), embedding in more than 2 dimensions is unsupported in both the official implementation (KlugerLab, 2020) and openTSNE (Poličar et al., 2019)

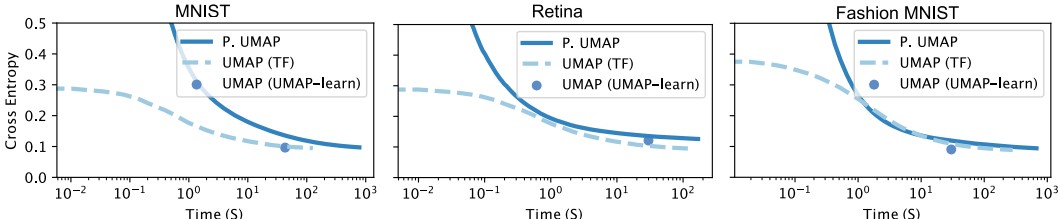

Figure 5: Training times comparison between UMAP and Parametric UMAP. All results were obtained with up to 32 threads on a machine with 2 AMD EPYC Rome 7252 8-Core CPU running at 3.1 GHz and a Quadro RTX 6000.

ethology, fast embedding methods like PCA or deep neural networks are necessary for real-time analysis and manipulations and deep neural networks are increasingly being used (e.g. Pandarinath et al. 2018; Brown & De Bivort 2018; Sainburg et al. 2019). Here, we compare the embedding speed of a held-out test sample for each dataset, as well as the speed of reconstruction of the same held out test samples.

Broadly, we observe similar embedding times for the non-parametric t-SNE and UMAP methods, which are several orders of magnitude slower than the parametric methods (Fig 6). Because the same neural networks are used across the different parametric UMAP and t-SNE methods, we show only Parametric UMAP in Fig 12, which is only slightly slower than PCA, making it a viable candidate for fast embedding in embedding paradigms where PCA is currently used.

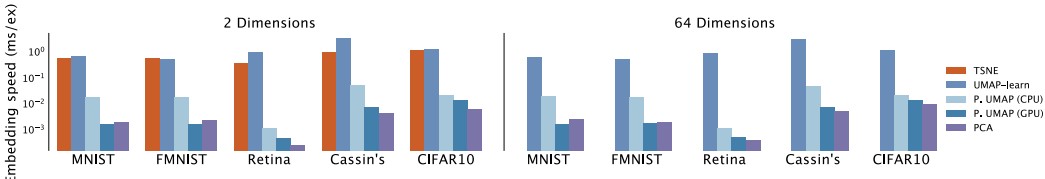

Figure 6: Comparison of embedding speeds using parametric UMAP and other embedding algorithms on a held-out testing dataset. Embeddings were performed on the same machine as Figure 11. Values shown are the median times over 10 runs.

Similar to the embedding speeds, we compared parametric and non-parametric UMAP reconstruction speeds (Fig 18). We find that the reconstructions of Parametric UMAP are orders of magnitude faster than non-parametric UMAP, and slightly slower, but within the same order of magnitude, as PCA.

## 2.3 AUTOENCODING

The ability to reconstruct data from embeddings can both aid in understanding the structure of nonlinear embeddings, as well as allow for manipulation and synthesis of data based on the learned features of the dataset. We compared the reconstruction accuracy across each method which had inverse-transform capabilities (i.e. $Z \rightarrow X$), as well as the reconstruction speed across the neural network-based implementations to non-parametric implementations and PCA. In addition, we performed latent space algebra on Parametric UMAP embeddings both with and without an autoencoder regularization, and found that reconstructed data can be linearly manipulated in complex feature space.

**Reconstruction accuracy** We measured reconstruction accuracy as Mean Squared Error (MSE) across each dataset (Fig 7; Table 7). In two dimensions, we find that Parametric UMAP typically reconstructs better than non-parametric UMAP, which in turn performs better than PCA. In addition, the autoencoder regularization slightly improves reconstruction performance. At 64 dimensions, the AE regularized Parametric UMAP is generally comparable to the AE and VAE and performs better than Parametric UMAP without autoencoder regularization. The non-parametric UMAP recon-

struction algorithm is not compared at 64 dimensions because it relies on an estimation of Delaunay triangulation, which does not scale well with higher dimensions.

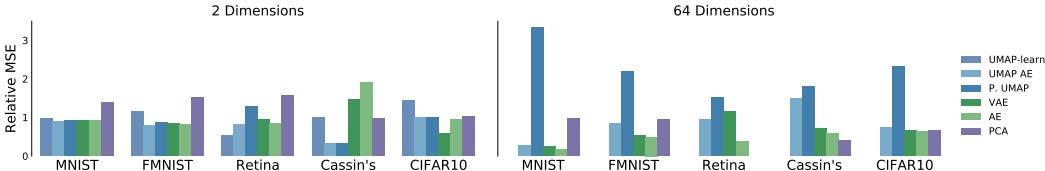

Figure 7: Reconstruction accuracy measured as mean squared error (MSE). MSE is shown relative to each dataset (setting mean at 1).

**Latent features**   Previous work shows that parametric embedding algorithms such as AEs (e.g. Variational Autoencoders) linearize complex data features in latent-space, for example the presence of a pair of sunglasses in pictures of faces (e.g. Radford et al. 2015; White 2016; Sainburg et al. 2018). Here, we performed latent-space algebra and reconstructed manipulations on Parametric UMAP latent-space to explore whether UMAP does the same.

To do so, we use the CelebAMask-HQ dataset, which contains annotations for 40 different facial features over a highly-structured dataset of human faces. We projected the dataset of faces into a CNN autoencoder architecture based upon the architecture defined in Huang et al. (2018). We trained the network first using UMAP loss alone (Parametric UMAP; Fig 8 right), and second using the joint UMAP and AE loss (Fig 8 center). We then fit an OLS regression to predict the latent projections of entire dataset using the 40 annotated features (e.g. hair color, presence of beard, smiling, etc). The vectors corresponding to each feature learned by the linear model were then treated as feature vectors in latent space, and added and subtracted from projected images, then passed through the decoder to observe the resulting image (as in Sainburg et al. 2018).

We find that complex latent features are linearized in latent space, both when the network is trained with UMAP loss alone as well as when the network is trained with AE loss. For example, in the last set of images in Figure 10B, a smile can be added or removed from the projected image by adding or subtracting its corresponding latent vector.

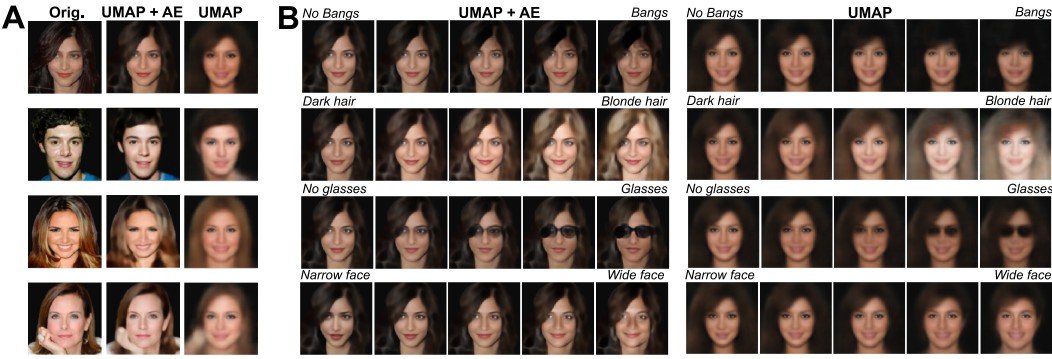

Figure 8: Reconstruction and interpolation. (A) Parametric UMAP reconstructions of faces from a holdout testing dataset. (B) The same networks, adding latent vectors corresponding to image features.

## 2.4   SEMI-SUPERVISED LEARNING

Real-word datasets are often comprised of a small number of labeled data, and a large number of unlabeled data. Semisupervised learning (SSL) aims to use the unlabeled data to learn the structure of the dataset, aiding a supervised learning algorithm in making decisions about the data. Current SOTA approaches in many areas of supervised learning such as computer vision rely on deep neural networks. Likewise, semisupervised learning approaches modify supervised networks with stucture-learning loss using unlabeled data. Parametric UMAP, being a neural network that learns structure

from unlabeled data, is well suited to semi-supervised applications. Here, we determine the efficacy of UMAP for semisupervised learning by jointly training a Y-shaped neural network (Fig 2D) on classification and UMAP compared to classification alone on datasets with varying numbers of labeled data.

We compared datasets ranging from highly-structured (MNIST) to unstructured (CIFAR10) in UMAP using a naïve distance metric in data space (e.g. Euclidean distance over images). For image datasets, we used a deep convolutional neural networks (CNN) which performs with relatively high accuracy for CNN classification on the fully supervised networks (see Table 8) based upon the CNN13 architecture commonly used in SSL (Oliver et al., 2018). For the birdsong dataset we used a BLSTM network, and for the retina dataset we used a densely connected network.

**Naïve UMAP embedding**   For datasets where structure is learned in UMAP (e.g. MNIST, FM-NIST) we expect that regularizing a classifier network with UMAP loss will aid the network in labeling data by learning the structure of the dataset from unlabeled data. To test this, we compared a baseline classifier to a network jointly trained on classifier loss and UMAP loss. We first trained the baseline classifier to asymptotic performance on the validation dataset, then using the pretrained-weights from the baseline classifier, trained a Y-shaped network (Fig 2D) jointly on UMAP over Euclidean distances and a classifier loss over the dataset. We find that for each dataset where categorically-relevant structure is found in latent projections of the datasets (MNIST, FMNIST, birdsong, retina), classifications are improved in the semi-supervised network over the supervised network alone, especially with smaller numbers of training examples (Fig 9; Table 8). In contrast for CIFAR10, the additional UMAP loss impairs performance in the classifier.

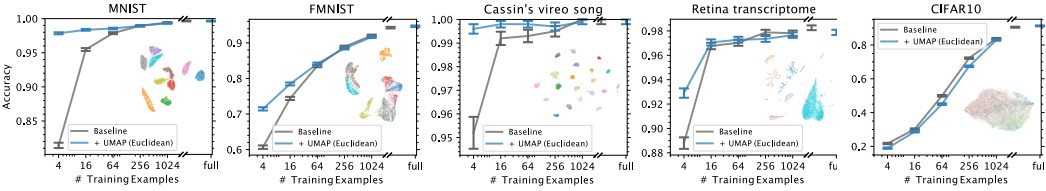

Figure 9: Baseline classifier with an additional UMAP loss with different numbers of labeled training examples. Non-parametric UMAP projections of the UMAP graph being jointly trained are shown in the bottom right of each panel. Error bars show SEM.

**Consistency regularization and learned invariance using data augmentation**   Several current SOTA SSL approaches employ a technique called consistency regularization (Sajjadi et al., 2016); training a classifier to produce the same predictions with unlabeled data which have been augmented and data that have not been augmented (Sohn et al., 2020; Berthelot et al., 2020). In a similar vein, for each image dataset, we train the network to preserve the structure of the UMAP graph when data have been augmented. We computed a UMAP graph over un-augmented data, and using augmented data, trained the network jointly using classifier and UMAP loss, teaching the network to learn to optimize the same UMAP graph, invarient to augmentations in the data. We observe a further improvement in network accuracy for MNIST and FMNIST over the baseline, and the augmented baseline (Fig 10; Table 8). For the CIFAR10 dataset, the addition of the UMAP loss, even over augmented data, reduces classification accuracy.

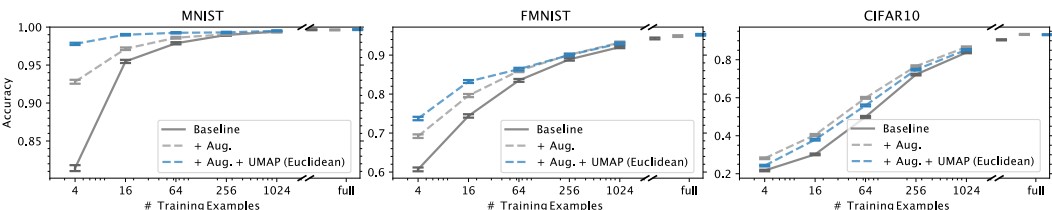

Figure 10: Comparison of baseline classifier, augmentation, and augmentation with an additional UMAP loss.

**Learning a categorically-relevant UMAP metric using a supervised network** It is unsurprising that UMAP confers no improvement for the CIFAR10 dataset, as UMAP computed over the pixel-wise Euclidean distance between images in the CIFAR10 dataset does not capture very much categorically-relevant structure in the dataset. Because no common distance metric over CIFAR10 images is likely to capture such structure, we consider using supervision to learn a categorically-relevant distance metric for UMAP. We do so by training on a UMAP graph computed using distance over latent activations in the classifier network (as in, e.g. Carter et al. 2019), where categorical structure can be seen in UMAP projection (Fig 19). The intuition being that training the network with unlabeled data to capture distributional-structure within the network's learned categorically-relevant space will aid in labeling new data.

We find that in all three datasets, without augmentation, the addition of the learned UMAP loss confers little no improvement in classification accuracy over the data (Fig 11; Table 8). When we look at non-parametric projections of the graph over latent space activations, we see that the learned graph largely conforms to the network's already-present categorical decision making (e.g. Fig 19 predictions vs. ground truth). In contrast, with augmentation, the addition of the UMAP loss improves performance in each dataset, including CIFAR10. This contrast in improvement demonstrates that training the network to learn a distribution in a categorically-relevant space that is already intrinsic to the network does not confer any additional information that the network can use in classification. Training the network to be invariant toward augmentations in the data, however, does aid in regularizing the classifier, more in-line with directly training the network on consistency in classifications (Sajjadi et al., 2016).

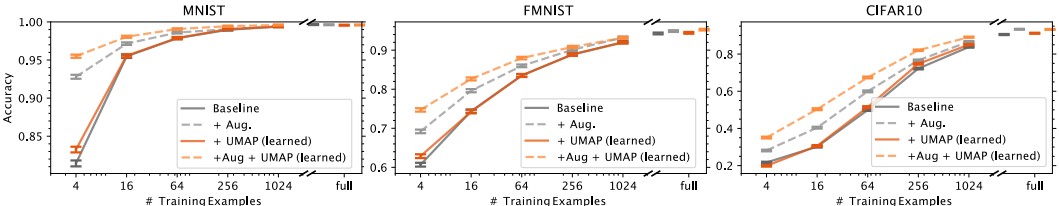

Figure 11: SSL using UMAP over the learned latent graph, computed over latent activations in the classifier.

## 3 DISCUSSION

In this paper we propose a novel parametric extension to UMAP using arbitrary deep neural networks. This parametric form of UMAP produces similar embeddings to non-parametric UMAP, with the added benefit of a learned mapping between data space and embedding space. We demonstrated the utility of this learned mapping on several downstream tasks. We showed that UMAP applied to deep neural networks improves inference times for embeddings and reconstructions by orders of magnitude while maintaining similar embedding quality to non-parametric UMAP. Combined with an autoencoder, UMAP improves reconstruction quality and allows for reconstruction of high-dimensional UMAP projections. We also show that parametric UMAP projections linearize complex features in latent space, similar to other latent neural networks such as AEs and GANs. Combined with a classifier, UMAP can be used for semi-supervised learning vastly improving training accuracy on datasets where small numbers of training exemplars are available. We showed that UMAP loss applied to a classifier improves semi-supervised learning in real world cases where UMAP projections carry categorically-relevant information (such as stereotyped birdsongs or single-cell transcriptomes), but not in cases where categorically relevant structure is not present (such as CIFAR10). We devised two downstream approaches based around learned categorically-relevant distances, and consistency regularization, that show improvements on these more complex datasets. Parametric embedding also makes UMAP feasible in fields where where dimensionality reduction of continuously generated signals plays an important role in real-time analysis and experimental control.

Future work can explore several of these topics such as on-line continuous embeddings of real-world datasets, semi-supervised learning with UMAP with improved categorically-relevant metrics, and applying UMAP loss to novel deep learning applications.

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

## A APPENDIX

### A.1 GRAPH CONSTRUCTION AND EMBEDDING

UMAP and t-SNE have the same goal: Given a $D$-dimensional data set $\mathbf{X} \in \mathbb{R}^D$, produce a $d$ dimensional embedding $Z \in \mathbb{R}^d$ such that points that are close together in $X$ (e.g. $x_i$ and $x_j$) are also close together in $Z$ ($z_i$ and $z_j$).

Both algorithms are comprised of the same two broad steps, first construct a graph of local relationships between datasets (Fig 1A), then optimize an embedding in low dimensional space which preserves the structure of the graph (Fig 1B). The parametric approach replaces the second step of this process with an optimization of the parameters of a deep neural network (Fig 1C).

#### A.1.1 GRAPH CONSTRUCTION

**Computing probabilities in $X$** The first step in both UMAP and t-SNE is to compute a distribution of probabilities $P$ between pairs of points in $X$ based upon the distances between points in data space. Probabilities are initially computed as local, one-directional, probabilities between a point and its neighbors in data-space, then symmetrized to yield a final probability representing the relationship between pairs of points.

In t-SNE, these probabilities are treated as conditional probabilities of neighborhood ($p_{i|j}^{\text{t-SNE}}$) computed using a Gaussian distribution centered at $x_i$.

$$p_{j|i}^{\text{t-SNE}} = \frac{\exp\left(-\mathrm{d}(\mathbf{x}_i, \mathbf{x}_j)/2\sigma_i^2\right)}{\sum_{k \neq i} \exp\left(-\mathrm{d}(\mathbf{x}_i, \mathbf{x}_k)/2\sigma_i^2\right)} \tag{1}$$

Where $\mathrm{d}(\mathbf{x}_i, \mathbf{x}_j)$ represents the distance between $x_i$ an $x_i$ (e.g. Euclidean distance) and $\sigma_i$ is the standard deviation for the Gaussian distribution, set based upon the a perplexity parameter such that one standard deviation of the Gaussian kernel fits a a set number of nearest-neighbors in $X$.

In UMAP, local, one-directional, probabilities ($P_{i|j}^{\text{UMAP}}$) are computed between a point and its neighbors to determine the probability with which an edge (or simplex exists), based upon an assumption that data is uniformly distributed across a manifold in a warped dataspace. Under this assumption, a local notion of distance is set by the distance to the $k^{\text{th}}$ nearest neighbor and the local probability is scaled by that local notion of distance.

$$p_{j|i}^{\text{UMAP}} = \exp(-(\mathrm{d}(\mathbf{x}_i, \mathbf{x}_j) - \rho_i)/\sigma_i) \tag{2}$$

Where $\rho_i$ is a local connectivity parameter set to the distance from $x_i$ to its nearest neighbor, and $\sigma_i$ is a local connectivity parameter set to match the local distance around $x_i$ upon its $k$ nearest neighbors (where $k$ is a hyperparameter).

After computing the one-directional edge probabilities for each datapoint, UMAP computes a global probability as the probability of either of the two local, one-directional, probabilities occurring $p_{ij} = \left(p_{j|i} + p_{i|j}\right) - p_{j|i}p_{i|j}$. In contrast, t-SNE symmetrizes the conditional probabilities as $p_{ij} = \frac{p_{j|i} + p_{i|j}}{2N}$.

#### A.1.2 GRAPH EMBEDDING

After constructing a distribution of probabilistically weighted edges between points in $X$, UMAP and t-SNE initialize an embedding in $Z$ corresponding to each data point, where a probability distribution ($Q$) is computed between points as was done with the distribution ($P$) in the input space. The objective of the UMAP and t-SNE is then to optimize that embedding to minimize the difference between $P$ and $Q$.

**Computing probabilities in $Z$** In embedding space, the pairwise probabilities are computed directly without first computing local, one-directional probabilities.

In the t-SNE embedding space, the pairwise probability between two points $q_{i|j}^{\text{t-SNE}}$ is computed in a similar manner to $p_{i|j}^{\text{t-SNE}}$, but where the Gaussian distribution is replaced with the fatter-tailed

Student's t-distribution (with one degree of freedom), which is used to overcome the 'crowding problem' (van der Maaten & Hinton, 2008) in translating volume differences in high-dimensional spaces to low-dimensional spaces:

$$q_{ij}^{\text{t-SNE}} = \frac{\left(1 + \|\mathbf{z}_i - \mathbf{z}_j\|^2\right)^{-1}}{\sum_{k \neq l} \left(1 + \|\mathbf{z}_k - \mathbf{z}_l\|^2\right)^{-1}} \tag{3}$$

UMAP's computation of the pairwise probability $q_{ij}^{\text{UMAP}}$ between points in the embedding space $Z$ uses a different family of functions:

$$q_{ij}^{\text{UMAP}} = \left(1 + a \|z_i - z_j\|^{2b}\right)^{-1} \tag{4}$$

Where $a$ and $b$ are hyperparameters set based upon a desired minimum distance between points in embedding space. Notably, the UMAP probability distribution in embedding space is not normalized, while the t-SNE distribution is normalized across the entire distribution of probabilities, meaning that the entire distribution of probabilities needs to be calculated before each optimization step of t-SNE.

**Cost function**    Finally the distribution of embeddings in $Z$ is optimized to minimize the difference betwee $Q$ and $P$.

In t-SNE, a Kullback-Leibler divergence between the two probability distributions is used, and gradient descent in t-SNE is computed over the embeddings $C_{\text{t-SNE}} = \sum_{i \neq j} p_{ij} \log p_{ij} - p_{ij} \log q_{ij}$.

In UMAP, the cost function is cross-entropy, also optimized using gradient descent $C_{\text{UMAP}} = \sum_{i \neq j} p_{ij} \log \left(\frac{p_{ij}}{q_{ij}}\right) + (1 - p_{ij}) \log \left(\frac{1 - p_{ij}}{1 - q_{ij}}\right)$.

**Attraction and repulsion**    Minimizing the cost function over every possible pair of points in the dataset would be computationally expensive. UMAP and more recent varients of t-SNE both use shortcuts to bypass much of that computation. In UMAP, those shortcuts are directly advantageous to batch-wise training in a neural network.

The primary intuition behind these shortcuts is that the cost function of both t-SNE and UMAP can both be broken out into a mixture of attractive forces between locally connected embeddings and repulsive forces between non-locally connected embeddings.

**Attractive forces**    Both UMAP and t-SNE utilize a similar strategy in minimizing the computational cost over attractive forces: they rely on an approximate nearest neighbors graph[4]. The intuition for this approach is that elements that are further apart in data space have very small edge probabilities, which can be treated as effectively zero. Thus, edge probabilities and attractive forces only need to be computed over the nearest neighbors, non-nearest neighbors can be treated as having an edge-probability of zero. Because nearest-neighbor graphs are themselves computationally expensive, approximate nearest neighbors (e.g. Dong et al. 2011) produce effectively similar results.

**Repulsive forces**    Because most data points are not locally connected, we do not need to waste computation on most pairs of embeddings.

UMAP takes a shortcut motivated by the language model word2vec (Mikolov et al., 2013) and performs negative sampling over embeddings. Each training step iterates over positive, locally-connected, edges and randomly samples edges from the remainder of the dataset treating their edge probabilities as zero to compute cross-entropy. Because most data points are not locally connected and have a very low edge probability, these negative samples are, on average, correct, allowing UMAP to sample only sparsely over edges in the dataset.

---

[4]UMAP requires substantially fewer nearest neighbors than t-SNE, which generally requires 3 times the perplexity hyperparameter (defaulted at 30 here), whereas UMAP computes only 15 neighbors by default, which is computationally less costly.

In t-SNE, repulsion is derived from the normalization of $Q$. A few methods for minimizing the amount of computation needed for repulsion have been developed. The first, is the Barnes-Hut tree algorithm Van Der Maaten (2014), which bins the embedding space into cells and where repulsive forces can be computed over cells rather than individual data points within those cells. Similarly, the more recent interpolation-based t-SNE (FIt-SNE; Linderman et al. 2017; 2019) divides the embedding space up into a grid and computes repulsive forces over the grid, rather than the full set of embeddings.

## A.2 Datasets

We performed experiments over several different datasets varying in complexity. The *Cassin's vireo song* dataset (Hedley, 2016b;a; Sainburg et al., 2019) consists of spectrograms of 20 of the most frequently sung elements elements of Cassin's vireo song (zero padded to 32 frequency by 31 time bins) produced by several individuals recorded in the Sierra Nevada mountains of California. Despite being recorded in the wild, the Cassin's vireo song is relatively low noise and vocal elements are highly stereotyped. *MNIST* is a benchmark handwritten digits dataset in 28x28 pixels (Kluger-Lab, 2010). *Fashion MNIST* (FMNIST) is a dataset of fashion items in the same format as Fashion MNIST designed to be a more difficult classification problem than MNIST (Xiao et al., 2017). *CIFAR10* is a natural image dataset (32x32x3 pixels) with 10 classes (Krizhevsky, 2009). CIFAR10 classes are much less structured than the other datasets used. For example, unlike FMNIST, subjects of images are not uniformly centered in the image and can have different background conditions that make neighborhood in pixel-space less likely between members of the same class. The *single-cell retina transcriptome* dataset consists of PCA projections (50D) of single-cell RNA transcriptome data from mouse retina (Macosko et al., 2015; Poličar et al., 2019). The *CelebAMask-HQ* dataset consists of cropped and aligned photographs of celebrity faces with 40 facial feature annotations (Lee et al., 2020; Liu et al., 2015) and label masks corresponding to face-landmarks. We removed the background of each image to make the task of learning a structured embedding simpler for the neural network. A further description of each dataset is given in Table 1 and a 2D projection of each dataset using each embedding algorithm is given in Figure 12.

| Dataset | Dim. | # Train/Valid/Test | Citation |
|---|---|---|---|
| **Moons** | 2 | 1K/NA/NA | Pedregosa et al. (2011) |
| **Bison** | 3 | 50K/NA/NA | Duhaime (2019) |
| **Cassin's vireo song** | 32x31 | 24.98K/1K/1K | Hedley (2016b;a) |
| **MNIST** | 28x28 | 50K/10K/10K | KlugerLab (2010) |
| **Fashion MNIST** | 28x28 | 50K/10K/10K | Xiao et al. (2017) |
| **CIFAR10** | 32x32x3 | 40K/10K/10K | Krizhevsky (2009) |
| **Mouse retina transcriptomes** | 50 | 30.33K/4.81K/10K | Macosko et al. (2015) |
| **CelebA-HQ (128)** | 128x128x3 | 28K/1K/1K | Lee et al. (2020); Liu et al. (2015) |

Table 1: Datasets used across all of the analyses and visualizations in the paper. NA refers to a split that was not used in the paper (e.g. no validation or testing set was used for the bison visualization in Fig 4B.)

## A.3 Embedding algorithms

Neural network architectures for the Parametric networks differ between datasets. MNIST, FMNIST, and CIFAR10 use convolutional neural networks. The Cassin's vireo dataset uses an LSTM encoder (and decoder for the autoencoder). The Retina dataset uses a 3-layer MLP with 100-neurons per layer. Parametric t-SNE and UMAP used the same neural network architectures and optimizer. For UMAP, the distance metric used for Cassin's vireo song is a dynamic-time warping (DTW) metric. The UMAP Autoencoder (UMAP AE) uses the same architecture of the Parametric UMAP and t-SNE implementations, combined with a corresponding decoder network. The encoder network and decoder network are jointly trained on a reconstruction and UMAP loss function. We additionally trained a decoder for the Parametric UMAP network, in which the encoder is trained only on the UMAP loss, and is not jointly trained on a reconstruction loss. For both t-SNE and Parametric t-SNE perplexity was left at its default value of 30 across datasets. We also left the degrees of

freedom for parametric t-SNE at $\alpha = d - 1$, where $d$ is the number of dimensions in the latent projection. Parametric embeddings (UMAP, t-SNE) are initialized with the same random neural network weights. Non-parametric algorithms use their corresponding default initializations.

## A.4    EMBEDDING METRICS

### A.4.1    TRUSTWORTHINESS

Trustworthiness (Venna & Kaski, 2006) is a measure of how much of the local structure of a dataset is preserved in a set of embeddings. Trustworthiness is quantified by comparing each datapoint's nearest neighbors in the original space, to its nearest neighbors in the embedding space:

$$T(k) = 1 - \frac{2}{nk(2n - 3k - 1)} \sum_{i=1}^{n} \sum_{j \in \mathcal{N}_i^k} \max(0, (r(i,j) - k)) \tag{5}$$

where $k$ is the number of nearest neighbors trustworthiness is being computed over, and for each sample $i$, $\mathcal{N}_i^k$ are the nearest $k$ neighbors in the original space, and each sample $j$ is $i$'s $r(i,j)^{\text{th}}$ nearest neighbor. We compared the Trustworthiness using Scikit-learn's default $k$ value of 5. Trustworthiness is scaled between 0 and 1, with 1 being more trustworthy.

### A.4.2    KNN CLASSIFIER

Related to trustworthiness, a k-Nearest Neighbor's (KNN) classifier is a supervised algorithm that classifies each unlabeled point based on its k-nearest labeled datapoints. We applied a KNN classifier with k=1 (Fig 14) and k=5 (Fig 15) to each dataset.

### A.4.3    SILHOUETTE SCORE

As opposed to Trustworthiness, which measures the preservation of local structure in a projection, the silhouette score (Rousseeuw, 1987) measures how 'clustered' a set of embeddings are, given ground truth labels. Silhouette score is computed as the mean silhouette coefficient across embeddings, where the silhouette coefficient is the distance between each embedding and all other embeddings in the same class, minus the distance to the nearest point in a separate class. Silhouette score is scaled between -1 and 1, with 1 being more clustered.

### A.4.4    CLUSTERING

To compare clustering directly across embeddings, we performed $k$-Means clustering over each latent projection and compared each embeddings clustering on the basis of the normalized mutual information (NMI) between clustering schemes. For each latent projection, $k$-Means was used to cluster the latent projection with $k$ (the number of clusters) varied between $\frac{1}{2} - 1\frac{1}{2}$ times the true number of categories in the dataset. The clustering was repeated five times per $k$. The best clustering was then picked on the basis of the silhouette score between the $k$ clusters and the projections (i.e. without reference to ground truth).

## A.5    RELATED WORK

Beyond Parametric t-SNE and Parametric UMAP, a number of recent parametric dimensionality reduction algorithms utilizing structure-preserving constraints exist which were not compared here. In this work, we chose not to compare these algorithms to Parametric UMAP because the focus of Parametric UMAP is on downstream applications, and not embedding quality, which we confirm in Section 2.1 is similar to non-parametric UMAP. This work is relevant to ours and is mentioned here to provide clarity on the current state of parametric topologically motivated and structure preserving dimensionality reduction algorithms.

Moor et al., (topological autoencoders; 2020) and Hoffer et al. (Connectivity-Optimized Representation Learning; 2019) apply an additional topological structure-preserving loss using persistent homology over mini-batches to the latent space of an autoencoder.    Jia et al., (Laplacian Autoencoders; 2015) similarly defines an autoencoder with a local structure preserving regularization.

Mishne et al., (Diffusion Nets; 2019) define an autoencoder extension based upon diffusion maps that constrains the latent space of the autoencoder. Ding et al., (scvis; 2018) and Graving and Couzin (VAE-SNE; 2020) describe VAE-derived dimensionality reduction algorithms based upon the ELBO objective. Duque et al (geometry-regularized autoencoders; 2020) regularize an autoencoder with the PHATE (Potential of Heat-diffusion for Affinity-based Trajectory Embedding) embedding algorithm (Moon et al., 2019). Szubert et al (ivis; 2019) and Robinson (Differential Embedding Networks; 2020) both make use of Siamese neural network architectures with structure-preserving loss functions to learn embeddings. Pai et al., (DIMAL; 2019) similarly uses Siamese networks constrained to preserve geodesic distances for dimensionality reduction.

## B FIGURES

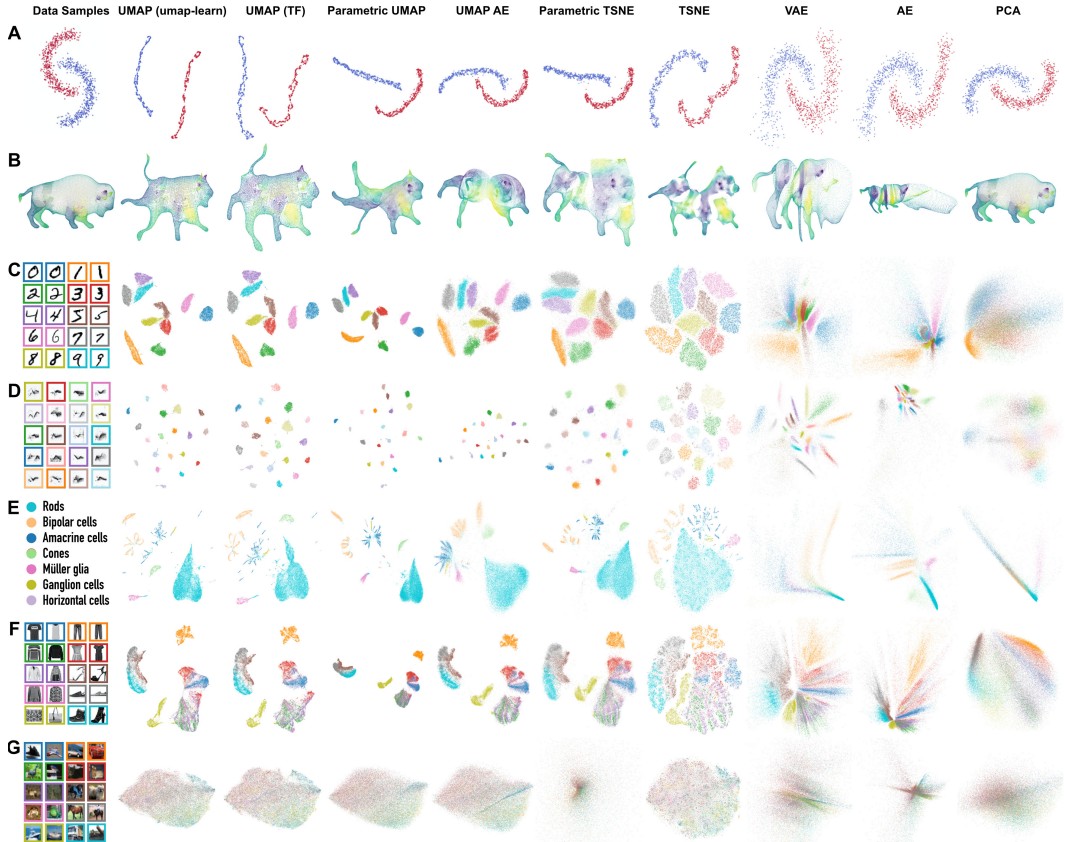

Figure 12: Comparison of projections from multiple datasets using UMAP, UMAP in Tensorflow, Parametric UMAP, Parametric UMAP with an Autoencoder loss, Parametric t-SNE, t-SNE, a VAE, an AE, and PCA. (a) Moons. (B) 3D buffalo. (c) MNIST (d) Cassin's vireo song segments (e) Mouse retina single cell transcriptomes. (f) Fashion MNIST (g) CIFAR10. The Cassin's vireo dataset uses a dynamic time warping loss and an LSTM network for the encoder and decoder for the neural networks. The image datasets use a convnet for the encoder and decoder for the neural networks. The bison examples use a t-SNE parplexity of 500 and 150 nearest neighbors in UMAP to capture more global structure.

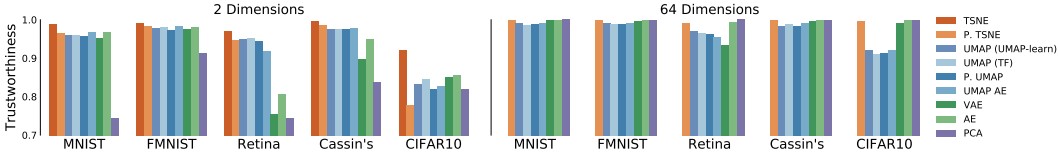

Figure 13: Trustworthiness scores for five datasets using 2- and 64-dimensional projections using each projection method. 64-dimensional t-SNE is not shown due to limitations in high-dimensional projections with t-SNE. Trustworthiness is computed over 10,000 samples of the training dataset. UMAP (TF) stands for the Tensorflow implementation of non-parametric UMAP.

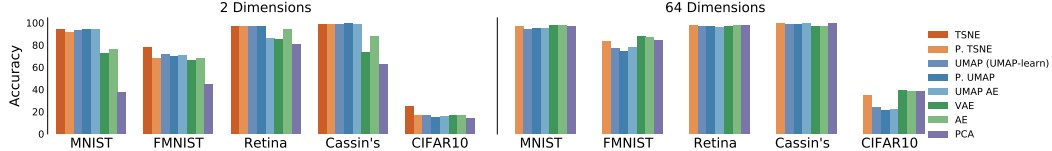

Figure 14: Generalization errors of KNN classifiers (k=1) on latent projections.

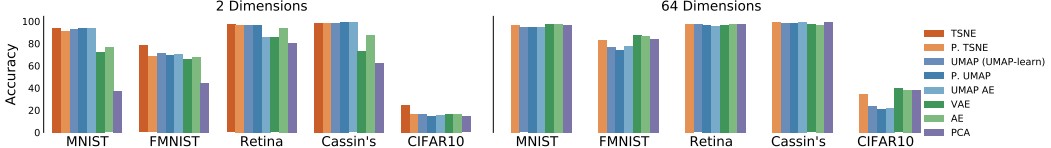

Figure 15: Generalization errors of KNN classifiers (k=5) on latent projections.

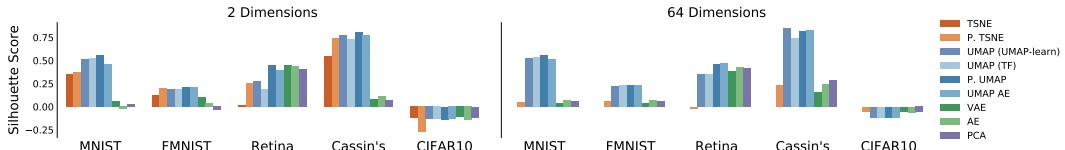

Figure 16: Silhouette scores for five datasets using 2- and 64-dimensional projections using each projection method. 64-dimensional t-SNE is not shown due to limitations in high-dimensional projections with t-SNE.

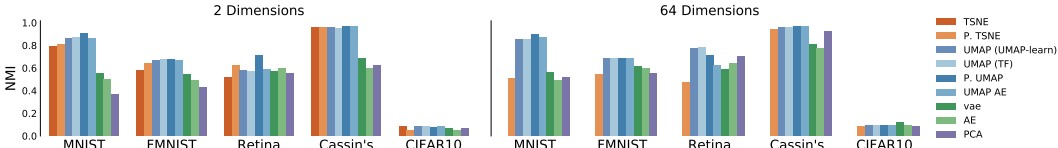

Figure 17: Clustering results. Comparisons are based upon the Normalized Mutual Information (NMI) between labels and clusters. Each dataset shows the NMI for the best clustering chosen on the basis of its silhouette score.

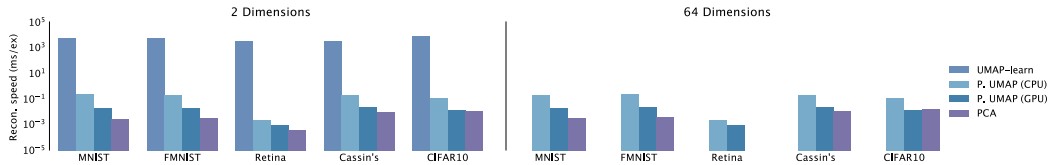

Figure 18: Reconstruction speed. Reconstructions are performed on the same machine as in Fig 6. Reconstructions are not shown for the retina dataset with PCA with a 64D latent space because the dataset is only 50 dimensions. Because all neural network architectures are held constant, speeds remain equal across each Parametric UMAP and t-SNE implementation. Values show the median time over 10 runs.

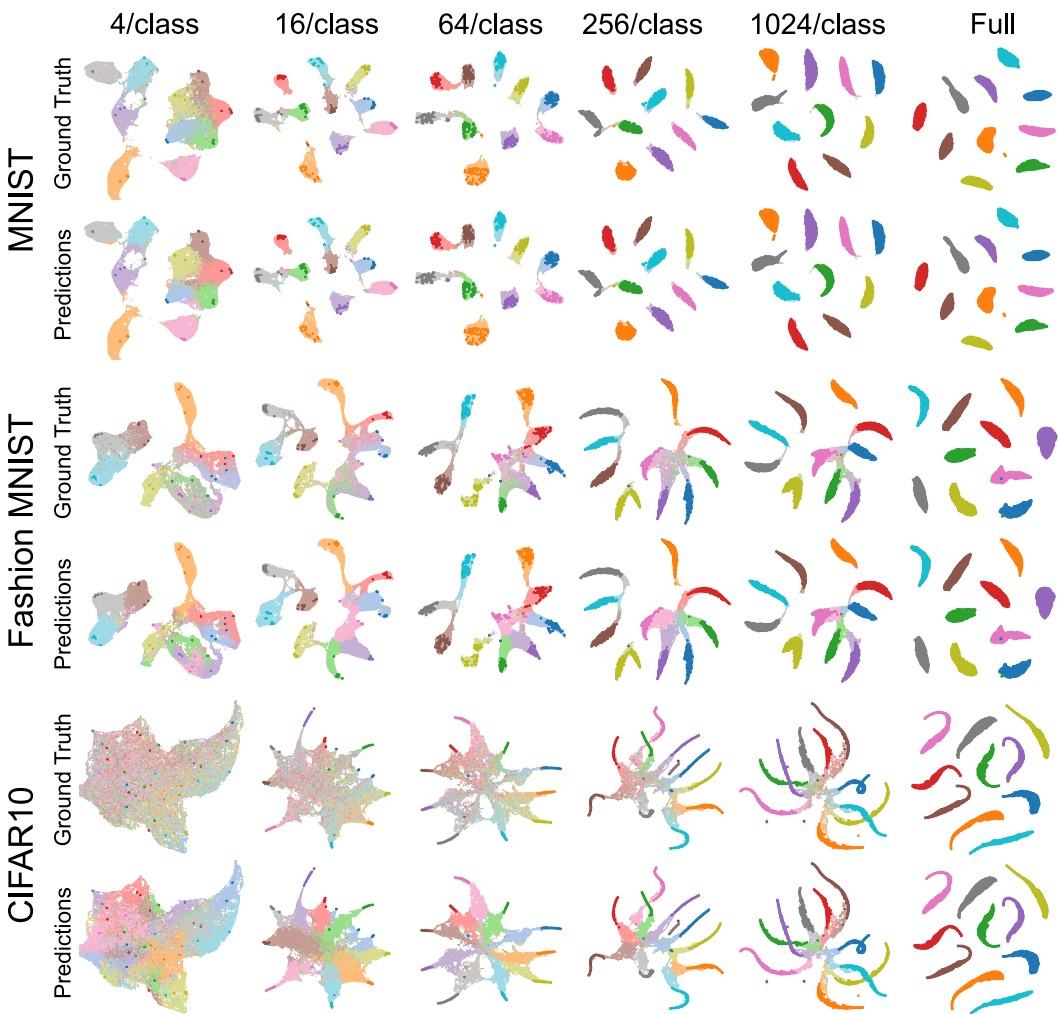

Figure 19: Non-parametric UMAP projections of activations in the last layer of a trained classifier for MNIST, FMNIST, and CIFAR10. For each dataset, the top row shows the ground truth labels on above, and the model's predictions below, in a light colormap. On top of each projection, the labeled datapoints used for training are shown in a darker colormap.

## B.1 RESULTS TABLES

| Dataset | Dim. | t-SNE | P. t-SNE | UMAP | P. UMAP | UMAP/AE | AE | VAE | PCA |
|---|---|---|---|---|---|---|---|---|---|
| Cassin's | 2 | **0.9949** | 0.9867 | 0.9758 | 0.9756 | 0.9777 | 0.9488 | 0.8976 | 0.8380 |
|  | 64 | - | 0.9990 | 0.9831 | 0.9840 | 0.9907 | 0.9981 | 0.9949 | **0.9999** |
| CIFAR10 | 2 | **0.9216** | 0.7773 | 0.8310 | 0.8187 | 0.8273 | 0.8564 | 0.8510 | 0.8202 |
|  | 64 | - | 0.9971 | 0.9209 | 0.9140 | 0.9199 | 0.9992 | 0.9913 | **0.9996** |
| FMNIST | 2 | **0.9906** | 0.9827 | 0.9777 | 0.9733 | 0.9842 | 0.9803 | 0.9751 | 0.9126 |
|  | 64 | - | 0.9991 | 0.9897 | 0.9894 | 0.9913 | 0.9991 | 0.9960 | **0.9995** |
| Retina | 2 | **0.9702** | 0.9463 | 0.9494 | 0.9435 | 0.9173 | 0.8063 | 0.7533 | 0.7445 |
|  | 64 | - | 0.9918 | 0.9708 | 0.9628 | 0.9542 | 0.9921 | 0.9342 | **1.0000** |
| MNIST | 2 | **0.9874** | 0.9655 | 0.9601 | 0.9573 | 0.9675 | 0.9663 | 0.9513 | 0.7434 |
|  | 64 | - | 0.9997 | 0.9895 | 0.9880 | 0.9905 | 0.9997 | 0.9994 | **0.9999** |

Table 2: Trustworthiness score for each method from Fig 13.

| Dataset | Dim. | t-SNE | P. t-SNE | UMAP | P. UMAP | UMAP/AE | AE | VAE | PCA |
|---|---|---|---|---|---|---|---|---|---|
| Cassin's | 2 | 0.9880 | 0.9860 | 0.9860 | **0.9910** | 0.9890 | 0.8740 | 0.7300 | 0.6260 |
|  | 64 | - | **0.9950** | 0.9850 | 0.9880 | 0.9940 | 0.9950 | 0.9800 | 0.9950 |
| CIFAR10 | 2 | **0.2457** | 0.1675 | 0.1689 | 0.1512 | 0.1592 | 0.1696 | 0.1665 | 0.1436 |
|  | 64 | - | 0.3426 | 0.2375 | 0.2139 | 0.2223 | 0.3790 | **0.3949** | 0.3829 |
| FMNIST | 2 | **0.7825** | 0.6834 | 0.7144 | 0.6941 | 0.7083 | 0.6816 | 0.6646 | 0.4467 |
|  | 64 | - | 0.8300 | 0.7682 | 0.7431 | 0.7772 | 0.8671 | **0.8747** | 0.8398 |
| Retina | 2 | **0.9717** | 0.9661 | 0.9665 | 0.9643 | 0.8581 | 0.9429 | 0.8545 | 0.8085 |
|  | 64 | - | **0.9772** | 0.9721 | 0.9683 | 0.9576 | 0.9750 | 0.9670 | 0.9759 |
| MNIST | 2 | **0.9411** | 0.9118 | 0.9317 | 0.9402 | 0.9403 | 0.7647 | 0.7241 | 0.3765 |
|  | 64 | - | 0.9697 | 0.9449 | 0.9518 | 0.9481 | 0.9748 | **0.9785** | 0.9707 |

Table 3: KNN (k = 1) scores for each method from Fig 14

| Dataset | Dim. | t-SNE | P. t-SNE | UMAP | P. UMAP | UMAP/AE | AE | VAE | PCA |
|---|---|---|---|---|---|---|---|---|---|
| Cassin's | 2 | 0.9910 | 0.9930 | 0.9890 | **0.9950** | 0.9930 | 0.9090 | 0.7740 | 0.6910 |
|  | 64 | - | 0.9950 | 0.9860 | 0.9910 | **0.9970** | 0.9930 | 0.9880 | 0.9920 |
| CIFAR10 | 2 | **0.2608** | 0.2017 | 0.1936 | 0.1722 | 0.1833 | 0.2007 | 0.1941 | 0.1503 |
|  | 64 | - | 0.3556 | 0.2694 | 0.2519 | 0.2477 | 0.3728 | **0.3777** | 0.3769 |
| FMNIST | 2 | **0.8039** | 0.7361 | 0.7608 | 0.7407 | 0.7561 | 0.7339 | 0.7161 | 0.5055 |
|  | 64 | - | 0.8479 | 0.8059 | 0.7878 | 0.8028 | 0.8756 | **0.8830** | 0.8568 |
| Retina | 2 | **0.9795** | 0.9766 | 0.9792 | 0.9761 | 0.8933 | 0.9647 | 0.8795 | 0.8429 |
|  | 64 | - | 0.9813 | 0.9801 | 0.9748 | 0.9661 | **0.9817** | 0.9770 | 0.9806 |
| MNIST | 2 | 0.9502 | 0.9378 | 0.9544 | **0.9614** | 0.9537 | 0.7926 | 0.7649 | 0.4201 |
|  | 64 | - | 0.9734 | 0.9538 | 0.9680 | 0.9654 | 0.9758 | **0.9791** | 0.9727 |

Table 4: KNN (k = 5) scores for each method from Fig 14

| Dataset | Dim. | t-SNE | P. t-SNE | UMAP | P. UMAP | UMAP/AE | AE | VAE | PCA |
|---|---|---|---|---|---|---|---|---|---|
| Cassin's | 2 | 0.5431 | 0.7439 | 0.7749 | **0.8013** | 0.7714 | 0.1125 | 0.0853 | 0.0731 |
| | 64 | - | 0.2299 | **0.8536** | 0.8173 | 0.8271 | 0.2411 | 0.1583 | 0.2914 |
| CIFAR10 | 2 | -0.1216 | -0.2757 | -0.1340 | -0.1359 | -0.1320 | -0.1436 | **-0.1114** | -0.1142 |
| | 64 | - | -0.0536 | -0.1166 | -0.1163 | -0.1172 | -0.0644 | **-0.0529** | -0.0580 |
| FMNIST | 2 | 0.1251 | 0.2013 | 0.1936 | **0.2139** | 0.2060 | 0.0427 | 0.1064 | -0.0331 |
| | 64 | - | 0.0543 | 0.2195 | **0.2315** | 0.2305 | 0.0655 | 0.0376 | 0.0618 |
| Retina | 2 | 0.0151 | 0.2578 | 0.2800 | **0.4519** | 0.3973 | 0.4394 | 0.4449 | 0.4009 |
| | 64 | - | -0.0214 | 0.3522 | 0.4652 | **0.4662** | 0.4289 | 0.3873 | 0.4188 |
| MNIST | 2 | 0.3498 | 0.3710 | 0.5186 | **0.5559** | 0.4637 | -0.0258 | 0.0627 | 0.0228 |
| | 64 | - | 0.0488 | 0.5276 | **0.5571** | 0.5166 | 0.0653 | 0.0431 | 0.0569 |

Table 5: Silhouette score for each method from Fig 16.

| Dataset | Dim. | t-SNE | P. t-SNE | UMAP | P. UMAP | UMAP/AE | AE | VAE | PCA |
|---|---|---|---|---|---|---|---|---|---|
| Cassin's | 2 | 0.9628 | 0.9605 | 0.9581 | **0.9686** | 0.9660 | 0.5984 | 0.7198 | 0.6029 |
| | 64 | - | 0.9434 | 0.9596 | **0.9665** | 0.9662 | 0.7501 | 0.7923 | 0.9019 |
| CIFAR10 | 2 | 0.0688 | 0.0383 | **0.0743** | 0.0719 | 0.0730 | 0.0258 | 0.0560 | 0.0605 |
| | 64 | - | 0.0570 | 0.0733 | 0.0742 | 0.0746 | 0.0574 | **0.0905** | 0.0599 |
| FMNIST | 2 | 0.5408 | 0.6248 | **0.6603** | 0.6594 | 0.6602 | 0.4818 | 0.5319 | 0.4221 |
| | 64 | - | 0.4680 | 0.6602 | 0.6618 | **0.6635** | 0.5541 | 0.5639 | 0.5244 |
| Retina | 2 | 0.5124 | 0.5912 | 0.5510 | **0.7112** | 0.5862 | 0.5695 | 0.5691 | 0.5522 |
| | 64 | - | 0.4682 | **0.7763** | 0.6356 | 0.5793 | 0.5897 | 0.5908 | 0.6696 |
| MNIST | 2 | 0.7704 | 0.7446 | 0.8375 | 0.7824 | **0.8460** | 0.4093 | 0.5467 | 0.3234 |
| | 64 | - | 0.4977 | 0.7747 | 0.7818 | **0.8701** | 0.4253 | 0.5617 | 0.5010 |

Table 6: Clustering score for each method from Fig 17

| Dataset | Dim. | UMAP | P. UMAP | UMAP/AE | AE | VAE | PCA |
|---|---|---|---|---|---|---|---|
| Cassin's | 2 | 0.0085 | **0.0028** | **0.0028** | 0.0163 | 0.0125 | 0.0082 |
| | 64 | - | 0.0034 | 0.0028 | 0.0011 | 0.0013 | **0.0008** |
| CIFAR10 | 2 | 0.0528 | 0.0369 | 0.0364 | 0.0344 | **0.0217** | 0.0370 |
| | 64 | - | 0.0300 | 0.0094 | **0.0080** | 0.0084 | 0.0084 |
| FMNIST | 2 | 0.0347 | 0.0266 | **0.0240** | 0.0244 | 0.0253 | 0.0461 |
| | 64 | - | 0.0241 | **0.0092** | 0.0054 | 0.0058 | 0.0104 |
| Retina | 2 | **0.0003** | 0.0008 | 0.0005 | 0.0005 | 0.0006 | 0.0010 |
| | 64 | - | 0.0005 | 0.0003 | **0.0001** | 0.0003 | - |
| MNIST | 2 | 0.0393 | 0.0374 | **0.0360** | 0.0369 | 0.0371 | 0.0557 |
| | 64 | - | 0.0313 | 0.0027 | **0.0016** | 0.0024 | 0.0090 |

Table 7: Reconstruction error on held-out testing set for each method.

|  |  | 4 | 64 | 256 | 1024 | full |
|---|---|---|---|---|---|---|
| MNIST | Baseline | 0.814 | 0.979 | 0.990 | 0.994 | 0.996 |
|  | + Aug. | 0.928 | 0.986 | 0.990 | 0.994 | 0.996 |
|  | + UMAP (Euclidean) | **0.978** | 0.986 | 0.990 | 0.993 | 0.996 |
|  | + UMAP (learned) | 0.832 | 0.979 | 0.990 | 0.994 | 0.996 |
|  | + Aug. + UMAP (learned) | 0.955 | 0.991 | **0.994** | **0.996** | 0.996 |
|  | + Aug. + UMAP (Euclidean) | **0.978** | **0.992** | 0.993 | 0.995 | **0.997** |
| FMNIST | Baseline | 0.607 | 0.835 | 0.889 | 0.920 | 0.943 |
|  | + Aug. | 0.692 | 0.860 | 0.901 | **0.932** | 0.949 |
|  | + UMAP (Euclidean) | 0.714 | 0.841 | 0.885 | 0.916 | 0.947 |
|  | + UMAP (learned) | 0.629 | 0.835 | 0.889 | 0.920 | 0.944 |
|  | + Aug. + UMAP (learned) | **0.747** | **0.880** | **0.908** | **0.932** | **0.952** |
|  | + Aug. + UMAP (Euclidean) | 0.737 | 0.864 | 0.900 | 0.930 | **0.952** |
| CIFAR10 | Baseline | 0.217 | 0.499 | 0.722 | 0.838 | 0.905 |
|  | + Aug. | 0.281 | 0.599 | 0.766 | 0.867 | 0.933 |
|  | + UMAP (Euclidean) | 0.190 | 0.450 | 0.674 | 0.829 | 0.913 |
|  | + UMAP (learned) | 0.199 | 0.515 | 0.748 | 0.850 | 0.912 |
|  | + Aug. + UMAP (learned) | **0.351** | **0.674** | **0.820** | **0.891** | **0.932** |
|  | + Aug. + UMAP (Euclidean) | 0.243 | 0.560 | 0.748 | 0.852 | **0.932** |
| Cassin's vireo | Baseline | 0.952 | 0.993 | 0.995 | **1.000** | **0.999** |
|  | + UMAP (Euclidean) | **0.996** | **0.998** | **0.997** | 0.999 | **0.999** |
| Retina | Baseline | 0.888 | 0.970 | **0.979** | **0.978** | **0.983** |
|  | + UMAP (Euclidean) | **0.929** | **0.973** | 0.973 | 0.977 | 0.979 |

Table 8: Classification accuracy across each dataset and method for different numbers of labeled training examples.

ACKNOWLEDGMENTS

Work supported by [witheld for anonymity]

