# OpenReview forum: "Parametric UMAP: learning embeddings with deep neural networks for representation and semi-supervised learning"
_ICLR.cc/2021/Conference — Reject_

### Official Review · AnonReviewer1 · 2020-10-19
**A great paper**

**Rating:** 9
**Confidence:** 5

**Review:**

In the manuscript, the authors introduce a parametric version of UMAP, replacing the original embedding optimization step with a deep learning solution detecting a parametric relationship between data and embedding. The novel approach compares favourably with the standard algorithm and, as a major contribution, defines a loss function that can be employed for other important applications such as constraining the latent distribution of autoencoders, and improving classifier accuracy for semi-supervised learning.

The paper is well written, complete and thoroughly detailed, both in the theoretical and the experimental section. The introduced material represents a significant advancement in the field, becoming a valuable resource for researchers in several areas.

A couple of notes:
- An application to one or more large real world dataset (e.g. single-cell sequencing, or weather radar data) would strengthen even more the authors’ claims and the paper’s impact, so I would suggest to include it, at least in the Appendix.
- Fig.3 in the Appendix is extremely useful to graphically explain the algorithm to a broader audience - I understand the page length limit, but I would strongly recommend to fit it in the main text.
- I would also suggest to include (maybe in the Appendix) a kind of “how-to” fully worked example to help researchers in optimising the use of novel algorithm in a data exploration pipeline
- I would point out (within the limitation of the anonimity requirement) the availability of the code for the algorithm

---

> ### Author Response · Authors · 2020-11-13
> **We thank the reviewer for their thoughtful review of our paper**
>
> We thank the reviewer for their thoughtful review of our paper. We appreciate the reviewer’s positive view, and hope that in addressing the issues described we have further improved the paper. In this comment, we provide a brief summary of the reviewer’s main points and our efforts to address them. We then give point by point responses in the following comments.
>
> Summary of the reviewer’s main points:
> 1. **An application to one or more large real-world datasets should be included.**
>    - *Summary of response: On the reviewer’s suggestion, we included a single-cell sequencing, as well as a bioacoustics dataset in the semisupervised learning section.*
> 2.  **Figure 3 should be included in the main text.**
>    - *Summary of response: We included a modified version of Figure 3 in the main text (to accommodate for room) and the full version in the appendix.*
> 3. **The reviewer suggests including a “how to” example to help researchers extend our code to their own pipelines.**
>    - *Summary of response: We added a link to an anonymous colab notebook walking through the main steps of the algorithm.*
>
>
> We hope that our additions properly address the reviewer’s request and suggestions.
>
> Thank you.

---

> > ### Author Response · Authors · 2020-11-13
> > **real world dataset**
> >
> > > An application to one or more large real world dataset (e.g. single-cell sequencing, or weather radar data) would strengthen even more the authors’ claims and the paper’s impact, so I would suggest to include it, at least in the Appendix.
> >
> > As the reviewer suggests, we used the mouse retinal single-cell transcriptome dataset as well as the Cassin’s vireo birdsong dataset to serve as real world datasets for the semi-supervised learning section.

---

> > ### Author Response · Authors · 2020-11-13
> > **Bring Fig 3 to the main text**
> >
> > > Fig.3 in the Appendix is extremely useful to graphically explain the algorithm to a broader audience - I understand the page length limit, but I would strongly recommend to fit it in the main text.
> >
> > As the reviewer suggests, we moved Fig 3 to the main text.

---

> > ### Author Response · Authors · 2020-11-13
> > **A "how to" example**
> >
> > > I would also suggest to include (maybe in the Appendix) a kind of “how-to” fully worked example to help researchers in optimising the use of novel algorithm in a data exploration pipeline.
> > I would point out (within the limitation of the anonimity requirement) the availability of the code for the algorithm
> >
> > As the reviewer suggests, we added an (anonymous) colab notebook walkthrough of the method.

---

### Official Review · AnonReviewer4 · 2020-10-28
**non-convincing results of the experiments**

**Rating:** 7
**Confidence:** 5

**Review:**

The authors propose a parametric version of UMAP by replacing sampling embeddings in the optimization of UMAP with directly learning weights of a neural network. The paper is very well and clearly written, but I have several significant concerns:

1. I don't see significant methodological novelty. Replacing embeddings with neural networks learning seems to be quite basic and straightforward. It is certainly a cherry on top of original UMAP, but I am not sure it could be counted as a separate method. The simplicity of methodology could be neglected, if the authors demonstrated significant improvement in their experiments, especially on downstream tasks.
2. A large part of the experiments is devoted to the comparison with tSNE, however it is not very clear why there is a lack of comparison with other parametric methods, such as Topological Autoencoders. Also not very clear why the authors mention these very relevant methods only in Appendix and not in the introduction in the beginning.
3. The performance of parametric UMAP achieves similar results to non-parametric UMAP, which is certainly nice, but also quite expected. Therefore, I would consider applications of parametric UMAP to other downstream tasks as a more significant and interesting contribution. However, experiments on this part are not convincing at all (especially on CIFAR10 dataset). Would be interesting to see the performance on some other datasets. Also, it would be very interesting to see confidence intervals for Figures 15, 16, 18.
4. In terms of speed I also don't see an improvement compared to non-parametric UMAP (TF). I see clear improvement compared to UMAP-learn version, but this as far as I understood due to a different implementation of the original UMAP and not in particular novelty of this paper.

After authors' response to revisions, I reconsidered my evaluation and updated the score.

---

> ### Author Response · Authors · 2020-11-13
> **We thank the reviewer for their thoughtful review of our paper**
>
> We thank the reviewer for their thoughtful review of our paper. In this comment, we provide a brief summary of the reviewer’s main points and our efforts to address them. We then give point by point responses in the following comments.
>
> Summary of the reviewer’s main points:
> 1. **Embedding comparisons should be made with other Parametric methods, such as Topological Autoencoders.**
>    - *Summary of response: As the reviewer mentions in other parts of their review, the focus of this paper is on downstream applications and not the learned embeddings per-se. Embedding comparisons are made to confirm similar embeddings to non-parametric baselines. Because algorithms such as Topological Autoencoders have already been compared to the same non-parametric baselines, and our methods do not confer a substantial improvement over those baselines, additional comparisons to e.g. Topological Autoencoders would likely yield little additional insight. To ensure the readers are aware of these related algorithms however, we expanded our discussion of related parametric methods, now referenced in the main text.*
> 2. **Downstream results are not convincing, especially on CIFAR10. It would be interesting to see performance on other datasets.**
>    - *Summary of response: We included CIFAR10 in our analysis to demonstrate a failure case -- that, in datasets where little categorically-relevant structure is found by UMAP (or t-SNE) downstream gains by including a UMAP loss are minimal without further adaptations. We now emphasize that in the main text. In addition, we took the reviewer’s advice and added two real-world datasets to this analysis (one bioacoustics, one single cell transcriptomes) to demonstrate that in real-world datasets in which UMAP does capture categorically-relevant structure, including UMAP loss in the classifier does substantially improve performance in SSL settings.*
> 3. **The reviewer asks for clarification on the speed improvements with Parametric UMAP over non-parametric UMAP.**
>    - *Inference for Parametric UMAP is ~2-3 orders of magnitude faster for Parametric UMAP than non-parametric UMAP (either tensorflow or UMAP learn). Optimization is slower for Parametric UMAP and non-parametric UMAP, but within the same order of magnitude.*
>
>
> We hope that our updates to downstream analyses (2), as well as clarifications on speed improvements (3) and further inclusion of related works (1) are sufficient to convince the reviewer that our proposed method provides a novel and impactful contribution to the field, and that the reviewer will update their score accordingly.
>
> Thank you.

---

> > ### Author Response · Authors · 2020-11-13
> > **Downstream applications**
> >
> > > I don't see significant methodological novelty. Replacing embeddings with neural networks learning seems to be quite basic and straightforward. It is certainly a cherry on top of original UMAP, but I am not sure it could be counted as a separate method. The simplicity of methodology could be neglected, if the authors demonstrated significant improvement in their experiments, especially on downstream tasks.”
> >
> > *Reviewer comment summary: Downstream applications do not show sufficient improvement in their experiments to consider this work a novel advancement in the field.*
> >
> > We appreciate the reviewer’s suggestion that we focus our manuscript on downstream applications, and the advantages of our approach, which we have done in the present revision.
> >
> > To reiterate the improvements that Parametric UMAP confers over non-parametric UMAP: Parametric UMAP speeds up inference over non-parametric UMAP by several orders of magnitude. Parametric UMAP provides reconstructions of data that are higher-quality, faster, and allow arbitrary dimensionality of data in comparison with non-parametric UMAP. Parametric UMAP also shows very promising, but not SOTA, results on semi-supervised classification tasks, including with real-world bioacoustics and single-cell sequencing datasets (added in this version as requested in your other comment). Classifier improvements hold true particularly when very few labels per class are available. We have updated the text in several places, in response to all four reviewers comments, to make these advances more clear.
> >
> > An additional role our paper takes is discussing why Parametric UMAP is straightforward in terms of t-SNE and Parametric t-SNE. With t-SNE, replacing embeddings with neural networks is non-trivial, as we discuss in the paper. UMAP better facilitates this, and, although the extension we propose is algorithmically a straightforward extension of UMAP, our work is the first to implement this extension, discuss why it works, and apply it to novel deep learning applications.
> >
> > We hope that the changes we’ve made addressing your following comments will persuade you of the importance and novelty of our work.
> >
> > **TLDR**: In this comment, we list several novel downstream applications of Parametric UMAP and discuss why Parametric UMAP is a substantial contribution to the field of representation learning.

---

> > ### Author Response · Authors · 2020-11-13
> > **Comparisons with related works**
> >
> > > A large part of the experiments is devoted to the comparison with tSNE, however it is not very clear why there is a lack of comparison with other parametric methods, such as Topological Autoencoders. Also not very clear why the authors mention these very relevant methods only in Appendix and not in the introduction in the beginning.
> >
> > *Reviewer comment summary: Experiments compare Parametric UMAP to UMAP (and other baselines: t-SNE, Parametric UMAP, VAE, AE, PCA), but not more recent parametric methods (e.g. Topological Autoencoders). Discussion of other parametric methods is left to the Appendix.*
> >
> > This is an important point that we are now much more explicit about in the current revision.
> >
> > First, we extended our discussion of other parametric methods, and added a reference to them in the introduction, as the reviewer requests.
> >
> > As to why we do not make comparisons on the learned embeddings:
> >
> > The main contribution of our paper is to provide a parametric extension to UMAP which enables downstream applications: faster embeddings/reconstructions, joint training with classifiers/autoencoders, and the facilitation of future downstream applications, like online learning and arbitrarily sized dataset embeddings.
> >
> > The embedding quality experiments we provided ensure that our results are of a similar quality to the original algorithms, facilitating downstream applications, not improving upon them. Our method is not intended to provide an improvement over e.g. UMAP or t-SNE’s embeddings, but to show that we get similar results. Because e.g. topological autoencoders already provide comparisons with UMAP and t-SNE, a comparison of embeddings with Parametric UMAP is unlikely to provide additional insights.  We now mention this in the manuscript.
> >
> > **TLDR**: The embedding quality experiments exist to confirm Parametric UMAP performs similarly to e.g. UMAP, to facilitate downstream work (e.g. speed, autoencoding, classification sections), not to demonstrate improvements. The mentioned work has already been compared to e.g. UMAP and t-SNE, so a comparison to Parametric UMAP likely provides little additional insight.

---

> > ### Author Response · Authors · 2020-11-13
> > **Additional datasets for SSL**
> >
> > > The performance of parametric UMAP achieves similar results to non-parametric UMAP, which is certainly nice, but also quite expected. Therefore, I would consider applications of parametric UMAP to other downstream tasks as a more significant and interesting contribution. However, experiments on this part are not convincing at all (especially on CIFAR10 dataset). Would be interesting to see the performance on some other datasets. Also, it would be very interesting to see confidence intervals for Figures 15, 16, 18.
> >
> > *Reviewer comment summary: Parametric UMAP’s downstream classification performs poorly on less structured datasets, like CIFAR10. Additional experiments should be performed.*
> >
> > We took the reviewer’s advice to add the performance on other real-world datasets in which UMAP captures categorically-relevant structure in the data (a bioacoustics dataset and a single-cell sequencing dataset). We show that in these datasets, the addition of a UMAP loss improves semi-supervised classification accuracy with small numbers of labels.
> >
> > We included CIFAR to exemplify a failure case, i.e. a dataset where learning local structure does not provide much categorically relevant information about the dataset. The other datasets vary on how much categorically relevant structure exists in the data (e.g. MNIST contains very much categorically relevant structure, FMNIST a moderate amount, and CIFAR little to none).
> >
> > In addition, we added confidence intervals to the classifier performance figures as requested.
> >
> > **TLDR**: We added real-world datasets to our analysis as the reviewer recommended. We show that our method performs well on semi-supervised classification with few labels in these datasets. These results apply to datasets in which UMAP captures categorically-relevant structure (like stereotyped birdsong syllables) but not datasets where UMAP does not capture categorically-relevant structure (like CIFAR10).

---

> > ### Author Response · Authors · 2020-11-13
> > **Clarification on speed improvements**
> >
> > > In terms of speed I also don't see an improvement compared to non-parametric UMAP (TF). I see clear improvement compared to UMAP-learn version, but this as far as I understood due to a different implementation of the original UMAP and not in particular novelty of this paper.
> >
> > *Reviewer comment summary: The reviewer asks for a clarification on the speed improvements given by Parametric UMAP.*
> >
> > The main speed contribution is in embedding and reconstruction, and not training time. The training time of Parametric UMAP is necessarily slower than non-parametric UMAP, as optimizing embeddings is generally faster than optimizing neural network weights (there are many more parameters to optimize in the parametric setting).
> >
> > For embedding and reconstruction, we achieve several orders of magnitude in speed up from non-parametric UMAP. We did not re-implement the embedding and reconstruction algorithms in the tensorflow (TF) version of non-parametric UMAP, but they should be roughly equivalent to the UMAP-learn version.
> >
> > We moved speed figures to the main text to make this more clear.
> >
> > **TLDR**: Embedding new data and reconstruction are sped up by ~2-3 orders of magnitude. For the initial optimization training a neural network is slower than optimizing embeddings directly, but on the same order of magnitude with our datasets. We moved the figures corresponding to these results in-line with the text for clarity.

---

### Official Review · AnonReviewer2 · 2020-10-29
**Rather naïve approach lacking sufficient justification and comparisons**

**Rating:** 4
**Confidence:** 5

**Review:**

**Update following discussion:**

Following the revision by the authors and the discussion with them, I am updating my score from 3 (Clear rejection) to 4 (OK, but not good enough - rejection). This reflects in great part the revision the authors made to have the main paper (limited to 8 pages) be self contain and present their main results, while using the appendices for complementary and technical information.

However, I still maintain the paper is not ready for publication in its current form. The extension of UMAP to implement the optimization via a neural network applied to input data rather than directly assigning coordinates is rather straightforward. The advantages it provides over UMAP in terms of natural inference on new data without the need for separate (more computationally intensive) out of sample extension method are a direct result of this neural network implementation, and they would be true not only for UMAP, but in fact for any method implemented in a "parametric" way via a neural network compared to nonparametric coordinate assignment. Similarly, allowing the addition of reconstruction or classification objectives in training is clearly a direct byproduct of this neural network implementation as well, and not unique to UMAP.

Therefore, an important question has to be asked here for whether the UMAP loss is indeed a good choice for a loss term to impose on networks, for example, to enable visualization or improve various tasks. The authors already look into this to some extent by comparing to parametric tSNE as one alternative approach, but there are many others, as I mention in the initial review, relying on constructions from topological data analysis and manifold learning - most, if not all, of which rely on some graph construction on the data and then ensuring the coordinates provided by a hidden layer in the network match the relations encoded in the graph, similar to the proposed UMAP loss term. How are reconstruction and classification affected by using such other regularizations compared to the UMAP one? Is inference speed the same for these other approaches? How does training speed compare between them? One can clearly expect some tradeoff between such properties and the geometric information encoded by different methods (UMAP and tSNE emphasize clusters, while other methods may emphasize other patterns), but this should be discussed and demonstrated clearly rather than just ignoring the vast amount of related work on parametric approaches to capturing intrinsic geometry in data.

Now, beyond the described lack of relevant comparisons for autoencoding and semi-supervised classification, even simply as a parametric implementation of UMAP (which would be a rather narrow scope, which is not very enticing as a motivation on its own),  I am not sure this work is sufficient to establish the presented approach. First, for the inference or embedding speed - this is essentially and out of sample extension task. As such, even if one insists on only comparing to UMAP-based methods, there are multiple OOS methods that can be used, such as Nystrom, geometric harmonics, etc. Some analysis of the tradeoff between extension quality and speed seems warranted here, but as I said previously - I think a comparison should also be provided to other parametric embedding methods beyond just OOS of UMAP (and tSNE for that matter). Second, as the authors clarified in discussion - their approach relies on the suitability of the UMAP loss to be incorporated directly in the network optimization, essentially comparing activations to the UMAP graph. However, an alternative approach presented in related work is to provide a loss term between activations and a UMAP embedding. This second approach is more general, since other embeddings can also be considered there, but also probably has some disadvantages (for example, the a priori fixed dimensionality, as the authors suggest). The differences between these two approaches should be addressed better in the manuscript, and importantly, since previous work exists already on the embedding loss approach, the authors should present a comparison establishing the benefits of the graph-based loss one, in addition to discussion regarding them.

To conclude, the idea behind this work seems reasonable, albeit rather straightforward since it's a reimplementation of the UMAP optimization. However, as it currently stands, I find it is not mature enough for publication and would need nonnegligible amount of work to properly position the contribution provided by this work compared to previous and related ones. I would like to encourage the authors to invest the time in adding such comparisons and clarifying not only how they are different from other methods, but also how they are better, and why choose UMAP to begin with as the basis for their proposed loss terms (compared to various other approaches - not just tSNE).

---

**Initial review:**
Before getting into the details of this work, I note that in my opinion it should have been desk rejected for violating the page limit base on the way it is written. The main 8 pages of the paper are far from being self contained, and regularly reference materials from the appendix as integral parts that are crucial for the presentation and understanding here. These include not only methodological illustrations, but also all results establishing the method. In fact, the main paper here does not show ANY result - it only describes the setting for getting them. ALL the figures and tables showing results appear solely in the appendix. If we are to ignore the appendices and only judge the paper based on these main eight pages, then there is no support, no results, and very little in the way of presenting the method here. If, on the other hand, we include the result figures as integral parts of the main paper (as they should be), then it clearly has significantly more than eight pages. Considering most papers submitted to this conference do  try to provide a coherent and relatively self-contained presentation of their work within the page limit, according to the guidelines of the conference, while only leaving technical and supporting details to the appendix, I believe it would be inappropriate to consider this work as meeting the conference page limit.

As for the work itself, this paper presents a rather naïve attempt to combine together the UMAP visualization with deep learning. It essentially proposes to consider the coordinates optimized by UMAP as resulting from a neural network applied to input data. Then, instead of adjusting directly these coordinates via the UMAP optimization, the method here continues to backpropagate the coordinate optimization through the network to provide a parametric model, via a feed forward neural network, that embeds the data in low dimensions while preserving the local neighborhood structure in the same sense that UMAP, tSNE and LargeVis do with their nonparametric approach. This neural network formulation can also naturally be extended to consider other loss terms, such as reconstruction loss of autoencoders or any predictive loss (classification, regression, etc.) enabling supervised visualization.

From a methodological perspective, this is a pretty straight-forward extension of the UMAP optimization, and does not indicate a clear advantage over it for the main task of unsupervised visualization or dimensionality reduction, neither in embedding quality or scalability. The authors show some interesting results (albeit only in the appendix and not in the main paper) on supervised visualization and out of sample extension speed, but these are not compared to relevant baselines that directly aim to address these tasks. Moreover, there is significant related work that is either ignored by the authors, or just mentioned in passing in the appendix without providing proper discussion and comparison with the proposed method. For example, in A.4, the authors mention topological autoencoders, connectivity-optimized representation learning, SCVIS, VAE-SNE, geometry regularized autoencoders, IVIS, and Differential Embedding Networks, but they do not compare their work to any of these, even though such comparison seems highly relevant here. More work that is completely ignored here includes, for example, Diffusion Nets (Mishne et al., 2015), Laplacian Autoencoders (Jia et al., 2015), DIMAL (Pai et al., 2019). Finally, briefly looking at Duque et al. (2020) cited here, while the main method there uses PHATE coordinates to regularize autoencoders, it seem they have also proposed the incorporation of UMAP loss terms in autoencoders, albeit only mentioned as somewhat of a sidenote together with tSNE regularization in their appendix. A discussion about the difference between these two approaches should be added to the main paper here, and it seems some comparison between them should also be presented to establish the advantages of the proposed approach here. Hence, even without the page limit argument, it does not seem the work presented here reaches the ICLR acceptance threshold without major revision to its presentation, discussion, and results. I must therefore recommend its rejection at this stage.

---

> ### Author Response · Authors · 2020-11-13
> **We thank the reviewer for their thoughtful review of our paper.**
>
> We thank the reviewer for their thoughtful review of our paper. In this comment, we provide a brief summary of the reviewer’s main points and our efforts to address them. We then give point by point responses in the following comments.
>
> Summary of the reviewer’s main points:
>
> 1. **Figures and Tables integral to understanding the paper were included in the appendix, violating the page limit.**
>    - *Summary of response: We moved the figures and tables into the main text, and a section of the introduction walking through the algorithms underlying UMAP and t-SNE to the appendix.*
> 2. **The paper presents a naive attempt to combine UMAP with deep learning.**
>    - *Summary of response: It appears as if there may be some discrepancy between our method, and the reviewers interpretation of our method. We updated the main text to increase clarity.*
> 3. **Embedding quality comparisons with a greater range of papers should have been provided.**
>    - *Summary of response: The intention of the embedding quality section of our paper was to show comparable quality between parametric UMAP and its non-parametric form in relation to common embedding algorithms (t-SNE, Parametric t-SNE, VAE, AE, and PCA). Parametric UMAP is not meant to improve upon these methods. It’s improvements lie in downstream applications (faster inference, acting as a regularization in classifier networks, etc). Because we have shown that parametric and non-parametric embeddings are similar, additionally comparing algorithms that have already between compared with the non-parametric form of UMAP are not likely to extend previous results.*
>
> We hope that moving the locations of the figures to the main text (1), alongside our explanations and updates for clarity in the text for (2) and (3 are sufficient for the reviewer to update their review accordingly.
>
> Thank you.

---

> > ### Author Response · Authors · 2020-11-13
> > **Figures and tables in appendix**
> >
> > > Before getting into the details of this work, I note that in my opinion it should have been desk rejected for violating the page limit base on the way it is written. The main 8 pages of the paper are far from being self contained, and regularly reference materials from the appendix as integral parts that are crucial for the presentation and understanding here. These include not only methodological illustrations, but also all results establishing the method. In fact, the main paper here does not show ANY result - it only describes the setting for getting them. ALL the figures and tables showing results appear solely in the appendix. If we are to ignore the appendices and only judge the paper based on these main eight pages, then there is no support, no results, and very little in the way of presenting the method here. If, on the other hand, we include the result figures as integral parts of the main paper (as they should be), then it clearly has significantly more than eight pages. Considering most papers submitted to this conference do try to provide a coherent and relatively self-contained presentation of their work within the page limit, according to the guidelines of the conference, while only leaving technical and supporting details to the appendix, I believe it would be inappropriate to consider this work as meeting the conference page limit.
> >
> > *Reviewer comment summary: The reviewer states that including figures and tables in the appendix, rather than the main text violates page limit requirements, since these figures and tables are necessary for comprehension of the main text.*
> >
> > We have taken the reviewer’s advice and moved as many key figures from the appendix to the main text as possible. To do so, we moved an algorithmic overview of UMAP and t-SNE to the appendix, replacing it with a summary and reference to the appendix for more information.
> >
> > This was only a minor change but we believe that it fixes the reviewer’s concerns about needing to refer to the appendix for comprehension. We hope the reviewer will update their given score to reflect the non-reliance upon the appendix in the revised version.
> >
> > **TLDR**: We moved several figures that were in the appendix to the main text. To fit them, we moved an algorithmic description of t-SNE and UMAP to the appendix, replacing that section with a summary.

---

> > > ### Comment · AnonReviewer2 · 2020-11-18
> > > **Resolved page limit violation**
> > >
> > > Thank you for rearranging the paper. I will go more thoroughly over it, but it does seem the main paper is reasonably self contained in its current form. I will update my score accordingly, although (as said in the initial review) the page limit was not my only concern. I will reflect on other concerns in separate comments.

---

> > ### Author Response · Authors · 2020-11-13
> > **No clear advantages to non-parametric UMAP**
> >
> > > As for the work itself, this paper presents a rather naïve attempt to combine together the UMAP visualization with deep learning. It essentially proposes to consider the coordinates optimized by UMAP as resulting from a neural network applied to input data. Then, instead of adjusting directly these coordinates via the UMAP optimization, the method here continues to backpropagate the coordinate optimization through the network to provide a parametric model, via a feed forward neural network, that embeds the data in low dimensions while preserving the local neighborhood structure in the same sense that UMAP, tSNE and LargeVis do with their nonparametric approach. This neural network formulation can also naturally be extended to consider other loss terms, such as reconstruction loss of autoencoders or any predictive loss (classification, regression, etc.) enabling supervised visualization. From a methodological perspective, this is a pretty straight-forward extension of the UMAP optimization, and does not indicate a clear advantage over it for the main task of unsupervised visualization or dimensionality reduction, neither in embedding quality or scalability.
> >
> > *Reviewer comment summary: The reviewer states that our approach is naive, training a neural network to learn the mapping between data and UMAP embeddings. The reviewer continues that our method does not confer additional advantages over current methods.*
> >
> > We agree that our method is a relatively straightforward extension of UMAP, but disagree that there are no clear advantages to our approach.
> >
> > To be clear, our work extends upon prior work by directly minimizing the UMAP cross entropy over neural network weights using stochastic gradient descent, an advantage that the UMAP algorithm enables that was not available in prior methods e.g. t-SNE. Both our implementation and contextualization of this in terms of UMAPs most popular alternative, t-SNE, and its parametric form. You cannot optimize t-SNE loss directly over a neural network with minibatches due to an important normalization step. In our work, we demonstrate that you can do so with UMAP, using negative sampling within the minibatch.
> >
> > Second, our approach confers several advantages over non-parametric UMAP. We improve inference speed by several orders of magnitude and provide a parametric mapping between data and embeddings. We combine UMAP with an autoencoder loss to improve reconstruction quality and enable UMAP reconstructions with high-dimensions, and we combine UMAP with a classifier to improve classifications in a semi-supervised setting with small numbers of labeled exemplars.
> >
> > **TLDR**: Our work extends UMAP by using a neural network to directly minimize the UMAP cross entropy loss. This method is novel when applied to neural networks, has downstream applications in deep learning, as we show.

---

> > ### Author Response · Authors · 2020-11-13
> > **Supervised visualization and inference speeds**
> >
> > > The authors show some interesting results (albeit only in the appendix and not in the main paper) on supervised visualization and out of sample extension speed, but these are not compared to relevant baselines that directly aim to address these tasks.
> >
> > *Reviewer comment summary: The reviewer notes that we present supervised visualization results and out of sample extension speed tasks in the appendix and not the main paper.*
> >
> > We would like to thank the reviewer for their interest in these results.
> >
> > To be clear, we did not present any work on supervised visualization in the main text or the appendix. The reviewer may be referring to Figure 19, which contains non-parametric UMAP visualizations of classifier projections from the semi-supervised networks. Those visualizations are to help explain the semi-supervised results, and are common practice (e.g. see “Activation atlas” on distill.pub).
> >
> > The inference speed section remains in the main text, and is now accompanied by an in-line figure, as requested by the reviewer.
> >
> > **TLDR**: We moved a figure of inference speed to the main text, to accompany the results section. We did not present any work on supervised visualization.

---

> > ### Author Response · Authors · 2020-11-13
> > **Related works**
> >
> > > Moreover, there is significant related work that is either ignored by the authors, or just mentioned in passing in the appendix without providing proper discussion and comparison with the proposed method. For example, in A.4, the authors mention topological autoencoders, connectivity-optimized representation learning, SCVIS, VAE-SNE, geometry regularized autoencoders, IVIS, and Differential Embedding Networks, but they do not compare their work to any of these, even though such comparison seems highly relevant here. More work that is completely ignored here includes, for example, Diffusion Nets (Mishne et al., 2015), Laplacian Autoencoders (Jia et al., 2015), DIMAL (Pai et al., 2019).
> >
> > *Reviewer comment summary: The reviewer suggests that we provide comparisons to additional baselines, which are presently discussed only in the appendix.*
> >
> > First, we added discussion of the three papers/methods not included in our related works section. In addition, we added mention to this section and these works in the main text.
> >
> > As to why we did not make comparisons between these methods in the embedding quality section. The intention of the embedding quality section of our paper is to show comparable quality between Parametric UMAP and its non-parametric form, not to show that our method improves over prior methods. The importance of our work lies in downstream applications (faster inference, acting as a regularization in classifier networks, etc).
> >
> > Because we have shown that parametric and non-parametric embeddings are similar, and the related works that the reviewer mentions already detail comparisons with non-parametric UMAP and t-SNE, additionally comparing these algorithms is not likely to provide any additional insights beyond what is present in those papers.
> >
> > **TLDR**: We expanded our discussion of related works, and explain that the relative embedding quality of UMAP (parametric or not) to these baselines is not a central question in our paper.

---

> > > ### Comment · AnonReviewer2 · 2020-11-18
> > > **Comparisons are still needed**
> > >
> > > Thank you for adding further related work discussion. However, I disagree with the assertion that comparison to other parametric methods is not needed here just because your main purpose is to provide a parametric version of the nonparametric UMAP. There are multiple methods for parametric embedding and visualization, using various loss terms, and many of them also have certain strengths in downstream tasks. If you want to establish the benefits of parametric UMAP you have to justify that the UMAP loss is a good design choice by showing that its optimization gives some benefits over other loss terms that can be used by parametric methods. This is especially the case when some of the previous work comparing to UMAP actually indicates that its nonparametric version is perhaps not ideal for faithfully capturing the structure of the data, so this would not align with your argument here.

---

> > > > ### Author Response · Authors · 2020-11-24
> > > > **Comparisons with additional related works**
> > > >
> > > > We appreciate the reviewer's request for comparisons with additional methods, and we agree that a complete review and comparison of each non-parametric embedding algorithm that we discuss in the related works section would give a clear picture of the landscape of topological data representation and the relative merits of each algorithm on downstream tasks.
> > > >
> > > > We ask the reviewer to consider that we have already made comparisons across 9 parametric and non-parametric algorithms, across 5 datasets, and two sets of dimensionalities (2 & 64), i.e. comparisons across ~90 sets of trained embeddings. For these, each of the parametric embeddings (Parametric UMAP, UMAP AE, Parametric t-SNE, AE, VAE) were trained in a well-controlled manner, holding constant network architecture and optimization methods.
> > > >
> > > > We agree that it would benefit the field to perform a well controlled comparison across each of the currently existing parametric topological embedding algorithms, and extend each of these algorithms in a similar manner toward downstream tasks, as the reviewer requests (e.g. adding a supervised classifier network to each network). To perform a comparison of the same fidelity as those present in our current analysis, we would need ensure that each implemented algorithm is trained under the same conditions.
> > > >
> > > > In consideration of your comment, we performed a review of the code implementations of each of the 10 topological parametric embedding algorithms reviewed in our related works section. We found that, while most (7/10) of the algorithms had code implementations available, only a subset had documentation explaining how to use the algorithm with your own datasets (2/10), and most code implementations differed in the ML framework that they were written in. Several of the methods had multi-step optimizations in which careful consideration would be needed to keep architectures and optimization methods constant. Extending these algorithms with e.g. semisupervised learning components for downstream comparisons, as the reviewer recommends, or even re-implementing them in our own framework would require substantial work.
> > > >
> > > > To facilitate future comparisons with our own work however, In our own work we implement a pip-installable package, with instructions for use with other datasets and network architectures, included sphinx documentation on usage of our implementation, as well as multiple code implementations both with custom training loops and high-level Keras optimization exemplifying how to modify and extend our algorithm. In addition, we now include a colab notebook walkthrough, as noted in an above comment, walking the reader through each step of the algorithm.
> > > >
> > > > We hope these steps make comparisons easier in future work.

---

> > ### Author Response · Authors · 2020-11-13
> > **Clarification of algorithm**
> >
> > > Finally, briefly looking at Duque et al. (2020) cited here, while the main method there uses PHATE coordinates to regularize autoencoders, it seem they have also proposed the incorporation of UMAP loss terms in autoencoders, albeit only mentioned as somewhat of a sidenote together with tSNE regularization in their appendix. A discussion about the difference between these two approaches should be added to the main paper here, and it seems some comparison between them should also be presented to establish the advantages of the proposed approach here.
> >
> > *Reviewer comment summary: The reviewer points out work by Duque et al (2020) and asks for clarification on the relationship between their work and ours.*
> >
> > It appears as if the reviewer may be interpreting our method as being equivalent to Duque et al, which trains networks to predict non-parametric UMAP embeddings from data, as opposed to training a network on UMAP’s loss. In their paper, they train a neural network to minimize the distance between embeddings learned via non-parametric UMAP:
> >
> > > *... penalizing the discrepancy between the latent representation and the embedding E previously learned by a manifold learning algorithm” (Duque et al., 2020)*
> >
> > In other words, Duque et al minimize the discrepancy between latent representations learned through a neural network and the embeddings previously learned by a manifold learning algorithm. We train the network’s weights on UMAP’s loss directly.
> >
> > To address this potential misunderstanding, as the reviewer suggests, we have modified and incorporated Figure 1 in the main text to outline the steps underlying the algorithm.
> >
> > We also added an anonymous colab notebook walkthrough of the steps of our method, which we hope helps readers in understanding our work.
> >
> > **TLDR**: The approach Duque et al take is very different from our own. We modified Figure 1, in the main text, for clarity.

---

> > > ### Comment · AnonReviewer2 · 2020-11-18
> > > **Why is your approach better than Duque et al? You need to establish your advantages, not just claim these are technically different approaches.**
> > >
> > > Thank you for clarifying the difference from that paper. I see your point. However, there is still an open question about whether (and why) it is better to go with the direct optimization of the UMAP loss you propose here compared to the indirect approach proposed there.
> > >
> > > It seems to me once the training is done, both methods would essentially have a network that computes UMAP-like coordinates and would enjoy the same inference speedup and reconstruction quality (if you include this loss term). Similarly, both approaches can pose a regularization for a classifier, and these would essentially be two implementation aiming to make a hidden layer similar to UMAP coordinates.
> > >
> > > The only difference seems to be whether you need to run UMAP separately as a preprocessing step or not, but this is a rather minor difference and one that is not necessarily disadvantageous, considering (as you mention) this allows other manifold learning algorithms to be plugged in there regardless of whether they can be optimized by SGD or not. Some empirical evidence needs to be presented to evaluate your approach here compared to that previously established one.

---

> > > > ### Author Response · Authors · 2020-11-24
> > > > **Establishing advantages**
> > > >
> > > > We appreciate the reviewer's persistence that we spell out the difference between these two methods clearly in the text explaining the advantages Parametric UMAP. We added an explanation in the text.
> > > >
> > > >  The difference between Parametric UMAP and training a neural network to predict a set of non-parametric embeddings, is that Parametric UMAP optimizes directly over the structure of the graph, with respect to the architecture of the network as well as additional constraints (like reconstruction error or the arcuitecture of a network). In contrast, training a neural network to predict non-parametric embeddings does not take additional constraints into account.
> > > >
> > > > Consider, for example, training Parametric UMAP with a linear model (e.g. a neural network with no hidden layers and a linear activation function), versus training the same linear model to predict the non-linear, non-parametric, embeddings. In the first case, the objective of the network is to come up with the best  linear embedding of the UMAP graph that it can. In the second case, information about the structure of the graph has been lost, thus this method cannot be optimized to learn a linear embedding of the data that best preserves the structure of the graph.
> > > >
> > > >    The issue remains when incorporating additional losses or using different network architectures. The discrepancy method is not optimizing the embedding of the graph with respect to additional constraints, it's just minimizing the distance between two sets of embeddings.
> > > >
> > > > Another way to think about this is that  the weighted graph is an intermediate topological representation (notably of no specific dimensionality) and is the *best* representation of the data under UMAP's assumptions. The process of embedding the data in a fixed dimensional space is necessarily a lossy one. Optimizing over the graph directly avoids this loss.

---

> > > > > ### Comment · AnonReviewer2 · 2020-11-24
> > > > > **Re: Establishing advantages**
> > > > >
> > > > > I agree with your point that directly optimizing the UMAP loss is different from optimizing a discrepancy between two sets of embedding. It may even intuitively make sense that there would be some advantage to direct optimization, as you suggest, since there is one less lossy step in the middle so more structure is preserved. However, this can and should be demonstrated with proper quantitative comparison, rather than argued hypothetically.
> > > > >
> > > > > As for the incorporation of other loss terms - I am not following this argument here. Why can't additional loss or regularization terms be added to another architecture? One can have an autoencoder, which has a reconstruction loss, with added discrepancy loss (softly) enforcing some geometric or topological structure (w.r.t. UMAP embedding in this case), and in addition also add an auxiliary classifier with some supervised loss, which can come from the embedding layer or any other hidden layer for that matter. This would have the same compositional capabilities as your parametric model.
> > > > >
> > > > > The only difference I see is how to derive the loss term relating to the quality of the dimensionality reduced coordinates, either in comparison to the graph or in comparison to another embedding. I don't disagree with your claim that using the graph is better, but I would like to see results properly verifying this.

---

### Official Review · AnonReviewer3 · 2020-11-03
**An interesting parametric approach to UMAP.**

**Rating:** 4
**Confidence:** 5

**Review:**

Response to authors:

The authors have largely responded well to my original concerns. However, after reading through the discussions with other reviewers, I agree with reviewer 2 that more work is required to make this publishable. In particular, this should include comparisons to the other methods suggested and justification of the use of the UMAP loss function. Given this, I have downgraded my score accordingly.

Original Review:
This paper presents a parametric approach for UMAP, a dimensionality reduction method. This area is of interest to the community as dimensionality reduction can be useful in a lot of different tasks such as visualization, semisupervised learning, etc. If my comments below can be addressed, I would be willing to increase my score.

Pros
The parametric version presented here appears to work well in the experiments given. The incorporation of the UMAP loss directly in a neural network as a regularization is also interesting.

Cons
Some of the results don't appear to have a corresponding figure or table, e.g. "Reconstruction accuracy" in section 3.3. These should be included.

UMAP tends to inherit some of the weaknesses of t-SNE as it tends to overemphasize local structure at the expense of global structure. In particular, it's been shown [R1] that UMAP and t-SNE are basically equivalent when using the same initialization. Could similar results be obtained by using the same initialization as UMAP instead of including the regularization term?

UMAP is traditionally used for visualization. Some of the applications presented (e.g. SSL) require/would benefit from higher dimensions than 2 or 3. t-SNE is known to be considerably slower for higher dimensions. Does UMAP inherit this problem? If so, that should be mentioned as a potential drawback.

All of the figures are given in the appendix. While this allows for more results, I think it would be a better paper if some of the figures were included in the main paper and some results were moved to the appendix.

The authors should verify that their references are as up to date as possible. For example, the PHATE paper should be updated to the Nature Biotechnology version (not bioRxiv).

[R1] Kobak and Linderman, https://www.biorxiv.org/content/10.1101/2019.12.19.877522v1

---

> ### Author Response · Authors · 2020-11-10
> **Quick clarification**
>
> Dear reviewer,
>
> Thank you for your review. We are preparing a full response to each point made along with updates to the manuscript, which will be posted soon.
>
> I hope that you could provide a quick clarification to this point:
>
> "UMAP tends to inherit some of the weaknesses of t-SNE as it tends to overemphasize local structure at the expense of global structure. In particular, it's been shown [R1] that UMAP and t-SNE are basically equivalent when using the same initialization. Could similar results be obtained by using the same initialization as UMAP instead of including the regularization term?"
>
> I agree with the remark, I'm just unsure of what results / initialization / regularization term the reviewer is referring to.  Parametric UMAP does not initialize embeddings, as the embeddings are learned projections through a neural network. The initial embeddings are projections through the weights of an untrained neural network. The same is true for Parametric t-SNE (i.e. here the two parametric networks have the same random weight initialization).  Could you clarify what results you are referring to? Or perhaps this answers your question?
>
> Cheers
>
> *edit 11/13*: We responded to the question in our main response thread.

---

> ### Author Response · Authors · 2020-11-13
> **We thank the reviewer for their thoughtful review of our paper.**
>
> We thank the reviewer for their thoughtful review of our paper. We appreciate the reviewer’s overall positive view, and hope that in addressing the issues described we have further improved the paper. In this comment, we provide a brief summary of the reviewer’s main points and our efforts to address them. We then give point-by-point responses in the following comments.
>
> Summary of the reviewer’s main points:
> 1. **Some results don’t appear to have a corresponding figure or table.**
>    - *Summary of response: Each result in the text now references a corresponding figure and table, many of which are now moved to the main text inline with the result.*
> 2. **The reviewer asks if differences in structure preservation are the result of differing initializations between models, as seen in non-parametric t-SNE and UMAP.**
>    - *Summary of response: Parametric UMAP (and Parametric t-SNE) uses a random initialization based upon the starting weights of a neural network. We further discuss the global structure preservation and the reliance upon a nearest-neighbors graph in the detailed comments below.*
> 3. **It should be clarified whether the speed of Parametric UMAP is affected by the dimensionality of the embedding.**
>    - *Summary of response: The speed is not affected by the dimensionality of the embedding. We emphasized this in the text.*
> 4. **All of the figures are in the appendix. It would be better if some were moved to the main text.**
>    - *Summary of response: we moved many of the figures in the appendix to the main text, making room by moving a section of the introduction giving a background on UMAP and t-SNE to the appendix.*
> 5. **The authors should ensure that references are up to date.**
>    - *Summary of response: We updated all relevant references.*
>
> We hope that our changes addressing the reviewers comments are sufficient for the reviewer to update their review.
>
> Thank you.

---

> > ### Author Response · Authors · 2020-11-13
> > **Corresponding figures/tables**
> >
> > > “Cons: Some of the results don't appear to have a corresponding figure or table, e.g. "Reconstruction accuracy" in section 3.3. These should be included.”
> >
> > *Reviewer comment summary: The reviewer requests that we ensure a corresponding figure/table exists for each result.*
> >
> > We added inline figures corresponding to results in as many places as possible. For the remaining results,  references to the corresponding tables and figures were included in the main text.
> >
> > **TLDR**: We moved some figures to the main text for readability, and ensured that all statistics have corresponding figures/tables that are properly referenced.

---

> > ### Author Response · Authors · 2020-11-13
> > **Initialization**
> >
> > > “UMAP tends to inherit some of the weaknesses of t-SNE as it tends to overemphasize local structure at the expense of global structure. In particular, it's been shown [R1] that UMAP and t-SNE are basically equivalent when using the same initialization. Could similar results be obtained by using the same initialization as UMAP instead of including the regularization term?
> > [R1] Kobak and Linderman, https://www.biorxiv.org/content/10.1101/2019.12.19.877522v1”
> >
> > *Reviewer comment summary: The reviewer requests that we ensure a corresponding figure/table exists for each result.*
> >
> > We thank the reviewer for asking for clarification, which we’ve added to the manuscript.
> > For parametric UMAP, the initial embeddings are projections through the weights of an untrained neural network. The same is true for Parametric t-SNE. In other words, in our work the two parametric networks have the same random weight initialization and corresponding embedding.
> >
> > We agree that non-parametric UMAP and t-SNE tend to emphasize local structure at the expense of global structure. This is largely due to the reliance of UMAP and t-SNE on a nearest-neighbors graph. In our implementation, we experimented with ways to better preserve global structure in UMAP and t-SNE but leave that for future work. Training over minibatches, for example, could allow one to compute and optimize upon batchwise edge probabilities in high-dimensional space from random batch samples. In principal this could better emphasize global structure, rather than relying on edge weights computed via the nearest neighbor graph alone.
> >
> > As an aside, it is also worth noting that while initializations are important, cost functions, which differ between UMAP and t-SNE, also have an important influence on capturing global structure. A discussion and analysis controlling for initialization is given in [1] (though not peer reviewed).
> >
> > [1] https://towardsdatascience.com/tsne-vs-umap-global-structure-4d8045acba17
> >
> > **TLDR**: In our work, Parametric UMAP and t-SNE use the same random initialization with our method. Preserving global structure better in Parametric UMAP would make for a good future project, optimizing over batch-wise computations of edge probabilities in high-dimensional space.

---

> > ### Author Response · Authors · 2020-11-13
> > **Dimensionality**
> >
> > > UMAP is traditionally used for visualization. Some of the applications presented (e.g. SSL) require/would benefit from higher dimensions than 2 or 3. t-SNE is known to be considerably slower for higher dimensions. Does UMAP inherit this problem? If so, that should be mentioned as a potential drawback.
> >
> > *Reviewer comment summary: The reviewer notes that UMAP and t-SNE are usually used for low dimensional projections. With t-SNE in particular, higher dimensional embeddings become intractable. The reviewer suggests that, if this is an issue for Parametric UMAP, we should note that.*
> >
> > To answer the question directly, this is not an issue for Parametric UMAP. Non-parametric UMAP also has no issues with embedding in high-dimensional space. We now mention this in the manuscript.
> > As an aside, this is also largely an advantage of Parametric t-SNE over non-parametric t-SNE, which minimizes t-SNE KL loss directly over subsets of the dataset using gradient descent, rather than relying on the optimization techniques reliant upon low-dimensional embeddings that non-parametric t-SNE uses (e.g. Barnes-Hut or Fourier interpolation based t-SNE).
> >
> > For reconstructing embeddings, Parametric UMAP confers a notable advantage over non-parametric UMAP. Non-parametric UMAP can only reconstruct data from low-dimensional embeddings (since it requires Delaunay triangulation). In contrast Parametric UMAP can reconstruct data of any dimensionality, similar to an autoencoder. This is why we don’t perform reconstructions on the 64D embeddings with non-parametric UMAP (Table 7).
> >
> > **TLDR**: We clarify that Parametric UMAP has no issues with dimensionality for embedding nor does non-parametric UMAP. It does confer advantage over non-parametric UMAP for reconstructing high-dimensional embeddings.

---

> > ### Author Response · Authors · 2020-11-13
> > **Figures in appendix to main text**
> >
> > > All of the figures are given in the appendix. While this allows for more results, I think it would be a better paper if some of the figures were included in the main paper and some results were moved to the appendix.
> >
> > *Reviewer comment summary: The reviewer would like us to move figures from the appendix to main text, where possible.*
> >
> > We have moved many of the figures important for understanding our paper to the main text. To accommodate this, we moved the section giving background on graph construction and embedding in UMAP and t-SNE to the appendix, replacing it with a summary. We refer the reader to the appendix for more information in the summary.
> >
> > **TLDR**: We moved several key figures to the main text.

---

> > ### Author Response · Authors · 2020-11-13
> > **Update references**
> >
> > > The authors should verify that their references are as up to date as possible. For example, the PHATE paper should be updated to the Nature Biotechnology version (not bioRxiv).
> >
> > *Reviewer comment summary: The reviewer asks us to verify up to date references.*
> >
> > We went through the references and updated them where appropriate.

---

### Decision · Program_Chairs · 2021-01-07
**Final Decision**

**Decision:**

Reject

**Comment:**

We thank the authors (and reviewers) for engaging in a detailed and constructive discussion, and providing a revised version of the paper after the initial round of reviews.

Regarding quality, the work is technically correct and the amount of experiments significant. However, as highlighted by reviewers 2 and 3, some important questions remain unanswered, in particular 1) more empirical evidence to support the claim that the UMAP loss is a relevant for neural networks, and 2) more comparison with existing approaches (beyond t-SNE).

Regarding clarity, the paper is overall clear and pleasant to read. However, after the revision round, all details about the proposed methods have been moved to the annex. While the initial version was criticized for the opposite reason (all experiments were in a annex), the balance may not be found yet; e.g., the equation for the UMAP loss, which is at the core of the paper, would certainly find its place in the main part of the manuscript for an ICLR paper.

The originality is the weakest aspect of the paper (besides the lack of comparison with related work). As mentioned by several reviewers, plugging the UMAP loss to a differentiable model is nowadays an idea that lacks originality. What would be important to justify that such a "straightforward" idea makes it to ICLR would be to demonstrate convincingly that it outperforms existing alternative approaches.

Finally, regarding the significance of the work, it is limited by the lack of thorough comparison with existing method. On the other hand, if the method is implemented in a fast and easy-to-use package, it may find its public as illustrated by the positive evaluation of Reviewer 1 from a potential user point of view.